# Self-rolling of vanadium dioxide nanomembranes for enhanced multi-level solar modulation

Xing Li[1,2,3,9], Cuicui Cao[4,5,9], Chang Liu[1,2,3], Wenhao He[6], Kaibo Wu[1,2], Yang Wang[1,2,3], Borui Xu ®[1,2], Ziao Tian ®[7], Enming Song[3,8], Jizhai Cui[1,2,3], Gaoshan Huang ®[1,2,3], Changlin Zheng[6], Zengfeng Di ®[7], Xun Cao ®[4,5] ✉ & Yongfeng Mei ®[1,2,3,8] ✉

Thermochromic window develops as a competitive solution for carbon emissions due to comprehensive advantages of its passivity and effective utilization of energy. How to further enhance the solar modulation ($\triangle T_{sol}$) of thermochromic windows while ensuring high luminous transmittance ($T_{lum}$) becomes the latest challenge to touch the limit of energy efficiency. Here, we show a smart window combining mechanochromism with thermochromism by self-rolling of vanadium dioxide ($VO_2$) nanomembranes to enhance multi-level solar modulation. The mechanochromism is introduced by the temperature-controlled regulation of curvature of rolled-up smart window, which benefits from effective strain adjustment in $VO_2$ nanomembranes upon the phase transition. Under geometry design and optimization, the rolled-up smart window with high $\triangle T_{sol}$ and $T_{lum}$ is achieved for the modulation of indoor temperature self-adapted to seasons and climate. Furthermore, such rolled-up smart window enables high infrared reflectance after triggered phase transition and acts as a smart lens protective cover for strong radiation. This work supports the feasibility of self-rolling technology in smart windows and lens protection, which promises broad interest and practical applications of self-adapting devices and systems for smart building, intelligent sensors and actuators with the perspective of energy efficiency.

The optimization of building energy efficiency considerably dominates the progress of sustainable urban development, as energy systems for lighting, heating, and air condition in buildings occupy approximately 40% of global energy consumption[1,2], in which window-related energy loss is hardly ignored[3]. Smart windows (SWs) are one of the promising approaches to modulate solar light transmittance for the operation efficiency. Thermochromic windows due to season feasibility, simple stimulation, stable molding and zero power consumption are one of

[1]Department of Materials Science & State Key Laboratory of ASIC and Systems, Fudan University, Shanghai 200438, People's Republic of China. [2]Yiwu Research Institute of Fudan University, Yiwu 322000 Zhejiang, People's Republic of China. [3]International Institute of Intelligent Nanorobots and Nanosystems, Fudan University, Shanghai 200438, People's Republic of China. [4]State Key Laboratory of High Performance Ceramics and Superfine Microstructure, Shanghai Institute of Ceramics, Chinese Academy of Sciences, Shanghai 200050, People's Republic of China. [5]Center of Materials Science and Optoelectronics Engineering, University of Chinese Academy of Sciences, Beijing 100049, People's Republic of China. [6]State Key Laboratory of Surface Physics and Department of Physics, Fudan University, Shanghai 200433, People's Republic of China. [7]State Key Laboratory of Functional Materials for Informatics, Shanghai Institute of Microsystem and Information Technology, Chinese Academy of Sciences, Shanghai 200050, People's Republic of China. [8]Shanghai Frontiers Science Research Base of Intelligent Optoelectronics and Perception, Institute of Optoelectronics, Fudan University, Shanghai 200438, People's Republic of China. [9]These authors contributed equally: Xing Li, Cuicui Cao. ✉e-mail: cxun@mail.sic.ac.cn; yfm@fudan.edu.cn

excellent candidates as a salient category of SWs for passive solar modulation in buildings[4–6]. As to thermochromic materials, vanadium dioxide (VO$_2$) stands out the most due to its low trigger temperature (close to room temperature) and high energy efficiency[7,8]. Hitherto, numerous progresses in improving the luminous transmittance ($T_{lum}$) and solar modulation ($\triangle T_{sol}$) have been made with VO$_2$-based planar structures, such as nanocomposite[9,10], porous[11,12], grid[13,14], biomimetic[15,16] and multifunctional coatings[17,18]. However, due to the lighting requirement of buildings with a high luminous transmittance and relative low energy consumption, it brings a challenge of SWs to further increase the $\triangle T_{sol}$ and compensate energy efficiency as well. This dilemma can be attributed to that the enhancement of visible transmittance of a VO$_2$-based planar SW will inevitably be accompanied by the sacrifice of the infrared blocking effect at high temperatures, which leads to a reduction of $\triangle T_{sol}$ [1].

Recently, three-dimensional (3D) thermochromic SWs have been demonstrated based on planar polymer composites, including kirigami-inspired[19] and photo-triggered SWs[20], which associate thermo-actuation with thermochromic materials to enhance both $\triangle T_{sol}$ and $T_{lum}$ simultaneously. These 3D SWs improved the difference of light transmittance between triggered and low temperatures (i.e., a larger $\triangle T_{sol}$) through reconfigurable shapes. However, these thermo-actuated SWs bring a problem, which is the balance of the thermal expansion temperature of applied smart polymer and the phase transition temperature of the thermochromic material (e.g., VO$_2$)[19,20]. Interestingly, VO$_2$ itself is an excellent material for thermo-actuators with high response speed, low trigger temperature, and large deformation[21–25]. Nevertheless, there are few efforts on a combination of thermo-actuation with thermochromic property due to the difficulty and complexity of their synergy in on-demand shape deformation and light transmission.

The micro/nanoscale rolled-up technology has become a reliable approach to design smart 3D structures from normal planar structures[26,27], which enables a fabrication dimension for devices with functional integration in e.g., on-chip photonics[28–30], nanorobots[31,32] and micro-capacitor[33,34]. Here, integrated rolled-up VO$_2$ SWs with high $T_{lum}$ and $\triangle T_{sol}$ are realized by the combination of mechanochromism and thermochromism. Under effective strain adjustment, the controllable curvatures of rolled-up SWs achieve maximum deformation upon the phase transition. Therefore, the rolled-up SWs can shift between rolled and unrolled (flat) statuses of nanomembranes (NMs) for synergy of shape deformation and light transmission due to the reversible actuation between low and triggered temperatures. By optimizing the initial curvature and rolling structures periodicity, the rolled-up SW reaches high $\triangle T_{sol}$ (42.14%) and $T_{lum}$ (61.01%), and successfully modulates the indoor temperature for energy management as to the alternating seasons. As a smart lens protective cover, the flat status (after triggered phase transition) of rolled-up SW reflects infrared for protection of the lens from high power radiation, and the rolled-up status barely affects the normal operation of the camera at low temperatures. This research proves that self-rolling can bring improvement to SWs and lens protection, and applies in various passive self-adapting devices and systems.

## Results

### Mechanism of rolled-up SW for intelligent light control

For building energy saving and user comfort, SW should have efficient indoor heat modulation and excellent lighting characteristics. Figure 1a, b is the schematic of a SW system based on rolled-up NMs trapped between two glass layers for intelligent light and heat control according to ambient temperatures. At low $\tau$, the NMs maintain a rolled-up status for exposing the underneath transparent substrate, which is proved by the SEM image of the rolled-up SW, where the colored areas are the rolled-up structure and the other part is the quartz (Fig. 1c). This makes them essentially transparent to the heat-generating components of sunlight (near-infrared, visible and ultraviolet light), which can, therefore, pass through the window and warm up the space on the other side. It also allows the user to view the outdoor scenery nearly unobstructed. When the temperature is above trigger temperature ($\tau_c$), the rolled-up NMs automatically unroll to flat status by strain change in the phase transition and completely cover the areas which are exposed in previous rolled-up status (Fig. 1d and Supplementary Movie 1). Here, the synergy of shape deformation and thermochromism of rolled-up VO$_2$ NM blocks the heat-generating components of sunlight, preventing them from passing through the window and limiting the temperature increase on the other side. Moreover, the sunlight irradiation through the rolled-up SW can be simply equivalent to normal incident light regardless of the incident angle, which is beneficial from the particularity of the rolled-up structure (Supplementary Fig. 1 and Note 1).

Through placing the rolled-up SW on the badge of Fudan University at different temperatures, the change in light transmittance is clearly visualized. In the rolled-up status, the SW presents advantages in color reproduction and clarity for framing (Fig. 1e). After the temperature raises to $\tau_c$, the rolled-up SW unrolls to a flat status, and the badge gets blurred (Fig. 1f). This exceptional performance of the rolled-up SW can be successfully realized thanks to beyond $50 \times 50$ microstructures simultaneously performing highly consistent collaborative work with large deformation. Therefore, it is necessary to deeply explore the essential reasons for the high performance of the rolled-up SW to achieve further optimization of solar modulation, such as phase transition properties, large deformation and structure design.

### Characterization of VO$_2$ NM lattice structure

The physical properties of the VO$_2$ NM, especially the strain change during the phase transition, are crucially important for the

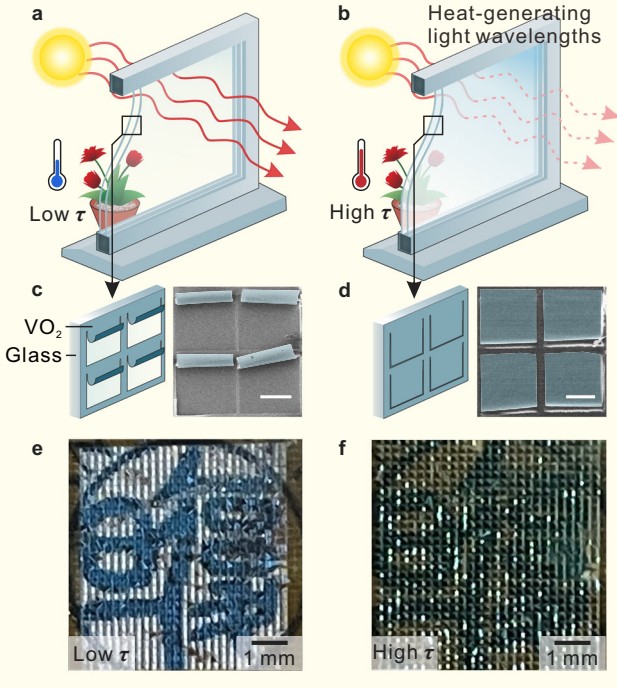

**Fig. 1 | Mechanism of rolled-up SW for intelligent light control. a** Schematic of rolled-up SW for high transparency at low temperatures (low $\tau$). **b** Schematic of rolled-up SW for blocking sunlight at high temperatures (high $\tau$). **c** Structure of rolled-up SW and corresponding the scanning electron microscopy (SEM) image at low $\tau$. Scaler bar: 100 μm. **d** Structure of rolled-up SW and corresponding the SEM image at high $\tau$. Scaler bar: 100 μm. **e** Photograph of rolled-up SW at low $\tau$. **f** Photograph of rolled-up SW at high $\tau$.

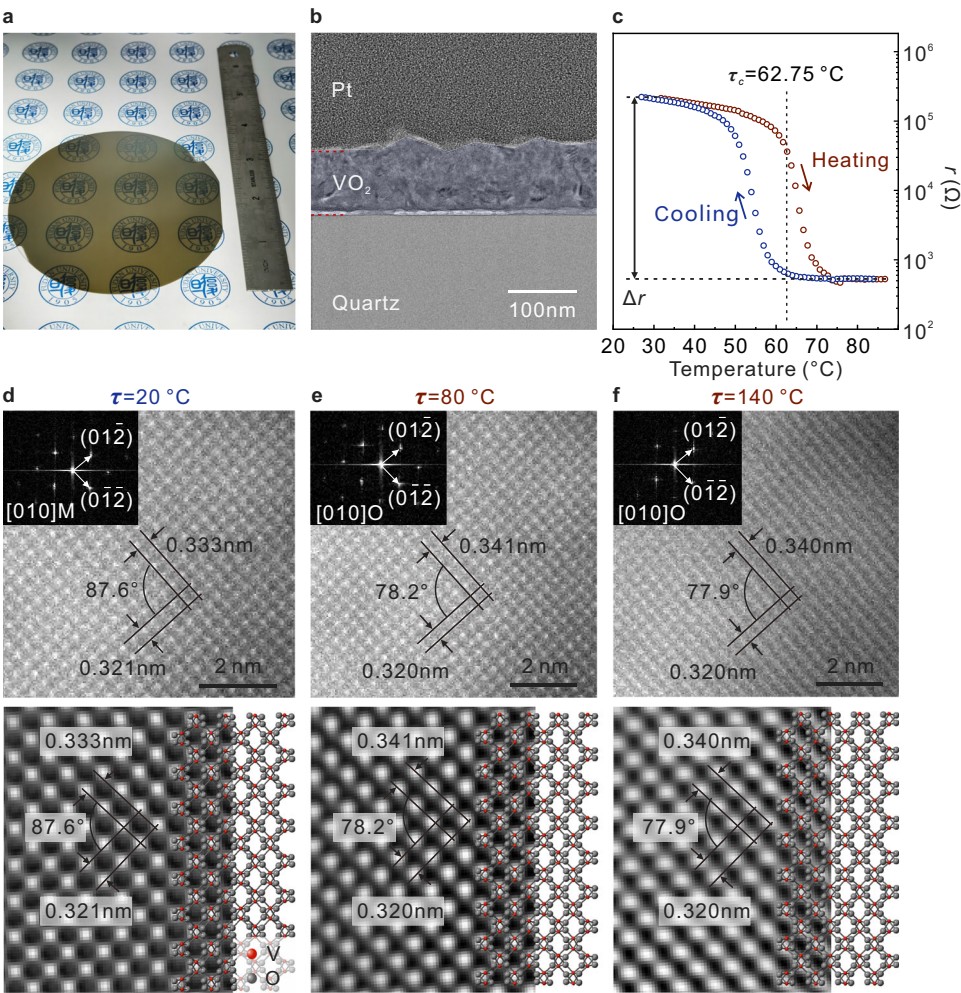

**Fig. 2 | Characterization of VO$_2$ NM lattice structure. a** The VO$_2$ NM grown on 4-inch quartz glass. **b** Transmission electron microscopy (TEM) image of the cross-section of VO$_2$ NM. **c** Temperature-dependent resistance (*r*) of VO$_2$ NM. △*r* corresponds to the change of resistance. **d–f** In situ scanning transmission electron microscopy (STEM) image and fast Fourier transformation (FFT) of VO$_2$ NM and corresponding inverse FFT image and crystal structure (lower region) **d** at 20 °C, **e** at 80 °C, and **f** 110 °C. Source data are provided as a Source Data file.

mechanochromism in the rolled-up SW. A VO$_2$ NM grown on a 4-inch wafer with good uniformity is shown in Fig. 2a. From the microscopic image shown by SEM, the surface morphology of the large scale VO$_2$ NM is flat and the crystal grain size is about 80 nm with uniform distribution (Supplementary Fig. 2), which facilitates uniform transmission of sunlight. In the transmission electron microscopy (TEM) image, the VO$_2$ NM between Quartz (lower part) and Pt (upper part. Pt layer mainly serves to protect the surface of the VO$_2$ NM during preparation of the cross-sectional sample.) possesses orientation in the vertical direction and a buffer layer is noticed near the substrate (Fig. 2b). Due to oxygen insufficiency during the initial growth stage, the buffer layer formed and was attributed to the formation of oxygen-deficient Magnéli phases V$_n$O$_{2n-1}$[35], but the buffer layer hardly affects the NM components and crystalline quality. The difference in components leads to a lattice mismatch between the upper pure VO$_2$ NM and buffer layer, which creates a stress gradient along the vertical direction for self-rolling (Supplementary Fig. 3). The high-resolution scanning transmission electron microscopy (STEM) images in different positions of the NM also show good crystallinity and similar crystal plane orientation (Supplementary Fig. 4). To further characterize the NM properties, the VO$_2$ NM was measured by small-angle synchrotron radiation XRD with different incidence angles ($\psi$) (Supplementary Fig. 5). According to these XRD results, the NMs grown with high temperature display four characteristic peaks commonly seen in VO$_2$

(M), which correspond to lattice planes of (011), (200), (210) and (220), respectively. With the increase of the $\psi$, the peak intensity of amorphous quartz around 12° decreases, and the intensity of VO$_2$ characteristic peak increases, while the peak position of VO$_2$ shifts to a smaller value. It indicates that the initial internal strain gradient is introduced into the VO$_2$ NM and it can be accurately calculated by a series of formulas (Supplementary Table 1, 2 and Note 2). For instance, the initial strain gradient of VO$_2$ NM deposited at 450 °C without annealing is about 0.5448%, which lies the foundation for the subsequent 3D structures assembly.

The resistance change in heating is a common measurement for judgment of the phase transition capability of VO$_2$. Here, the measurement results of temperature-dependent resistance (*r*) show that the change of resistance △*r* of a VO$_2$ NM is close to 2.1 × 10$^5$ Ω, ratio △*r*/*r* is 403.69 (*r* is resistance at high $\tau$), and $\tau_c$ is 62.75 °C (Fig. 2c). To expand the characterization of the NM quality across the wafer, the semiconductor characterization system was used to test the sample at various positions to extract the △*r*/*r* as well as its uniformity across the 4-inch wafer. More than 89% of the wafer area have the △*r*/*r* > 350, demonstrating high yield, good uniformity, and excellent quality of wafer-scale VO$_2$ NM (Supplementary Fig. 6). The variation of VO$_2$ physical properties during phase transition usually originates from the change of its lattice structure. In order to figure out the strain

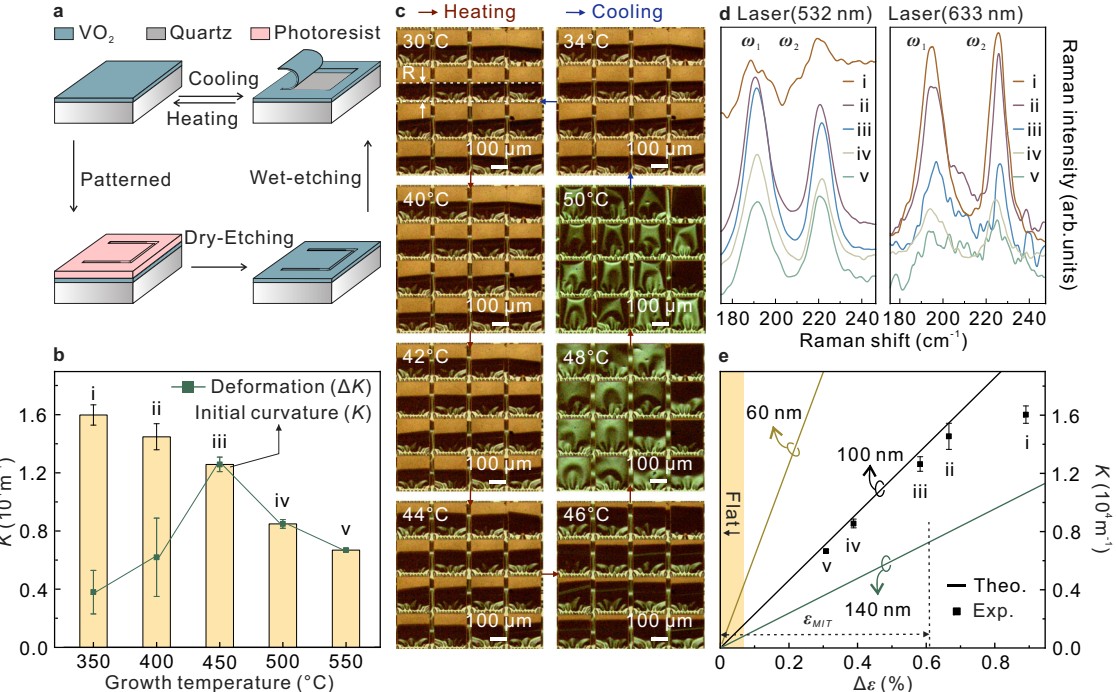

**Fig. 3 | Strain and deformation of rolled-up SW. a** Fabrication process of the rolled-up SW. **b** Initial curvature (K) of rolled-up SW with different growth temperatures and corresponding deformation (△K) during heating. i (350 °C), ii (400 °C), iii (450 °C), iv (500 °C), and v (550 °C). The corresponding error bar represents standard deviation. For calculation, different rolled-up structures on each sample were measured, and the numbers (n) of the measured rolled-up structures are $n_i = 1598$, $n_{ii} = 982$, $n_{iii} = 795$, $n_{iv} = 690$, and $n_v = 682$ (one measurement for each rolled-up structure). **c** Microscope image of sample iii at different temperatures. **d** Raman spectra of two low-frequency phonons $\omega_1$ and $\omega_2$ in rolled-up SW samples excited by lasers with wavelengths of 532 nm (left) and 633 nm (right). **e** Summary of rolled-up SW curvature (K) as a function of initial strain gradient (△ε). The solid lines represent the simulation results of the rolled-up SWs with different NM thicknesses and the solid points represent different rolled-up SW sample. The error bar corresponds to (**b**). Source data are provided as a Source Data file.

generation from metal-insulator transition (MIT), the STEM image of the VO₂ NM lattice structure was obtained at different temperatures. The atomic-resolution STEM image of VO₂ NM and corresponding fast Fourier transformation (FFT) are concordant with the [010] zone axis of the monoclinic structure at 20 °C (Fig. 2d). The bright dots with similar intensity in STEM images represent projected V atoms. According to the intensity of V atomic column, the interplanar spacing (d) in different zone axis is shown in the STEM image and an angle of 87.6° between different crystalline planes is noticed. In order to resolve the accurate atomic arrangement and crystal information, we built a crystal model based on the crystal file (mp-102155) and found that the model fits well with the inverse FFT image in lower region of Fig. 2d. It indicates that the bright dots in the STEM image is the overlap of two V atoms, but the values of d can still be trusted due to the uniform distribution between the V atoms. As the temperature increases to 80 °C, the number of bright spots in the FFT decrease, while the d in different zone axis has significant changes, and the angle between two crystalline planes changes to 78.2° (Fig. 2e). These crystal parameters in 80 °C correspond to the metallic O phase of VO₂ (mp-1094031)[36] which proves the completion of the phase transition. After the temperature is further increased to 140 °C, the FFT and d in Fig. 2f are almost unchanged compared to Fig. 2e, indicating that the change of crystal structure during heating is mainly attributed to the occurrence of phase transition and less affected by the increase of temperature after MIT. More STEM images taken at different temperatures are shown in Supplementary Fig. 7 and d values are labeled therein. According to the relationship between d and strain, a compressive strain generation during MIT, $\varepsilon_{MIT}$ (0.61%) has been calculated (Supplementary Note 2), which contributes to

the study of accurate control of the deformation of the rolled-up SW.

## Strain and deformation of rolled-up SW

To study the relationship between strain and deformation, rolled-up SW samples with different initial curvatures (K) were fabricated. There are 4 steps of the whole fabrication process for rolled-up SW (Fig. 3a). Firstly, the VO₂ NMs were deposited at different growth temperatures for various initial strains. The second step was patterning the rolled-up region by photolithography and the next step was etching the graphic edge by reactive-ion etching (RIE). Finally, HF solution was used to etch the quartz, which released the VO₂ NMs for rolling process. In order to fix the framework (i.e., un-rolled) regions, photoresist layer might be used to cover these regions, which helps to maintain the intact framework geometry (Supplementary Fig. 8). In specific experiments, the NM thickness, which has a strong influence on both light transmission and K of rolled-up structure, needs to be kept constant, and thus it is more conducive to the quantitative study of the strain-deformation relationship. Based on the light transmittance measurement results, 100 nm VO₂ NM had the best $\triangle T_{sol}$ during phase transition (Supplementary Fig. 9). Such a thick NM can be rolled into a tubular structure with large K, helping a further improvement of light transmittance at low τ. Control the NM at the same thickness by the same deposition time, the rolled-up SW samples deposited at different temperatures (i, ii, iii, iv, and v) demonstrate various initial K and deformations △K in MIT (Fig. 3b, detailed experimental results in Supplementary Figs. 10–14 and calculation method of error bar is standard deviation). The microscope images during heating and K as a function of temperature for different samples are respectively shown in Supplementary Fig. 15. As the growth temperature raises, K of the rolled-up SW reduces significantly, which leads to a decreased area of quartz directly

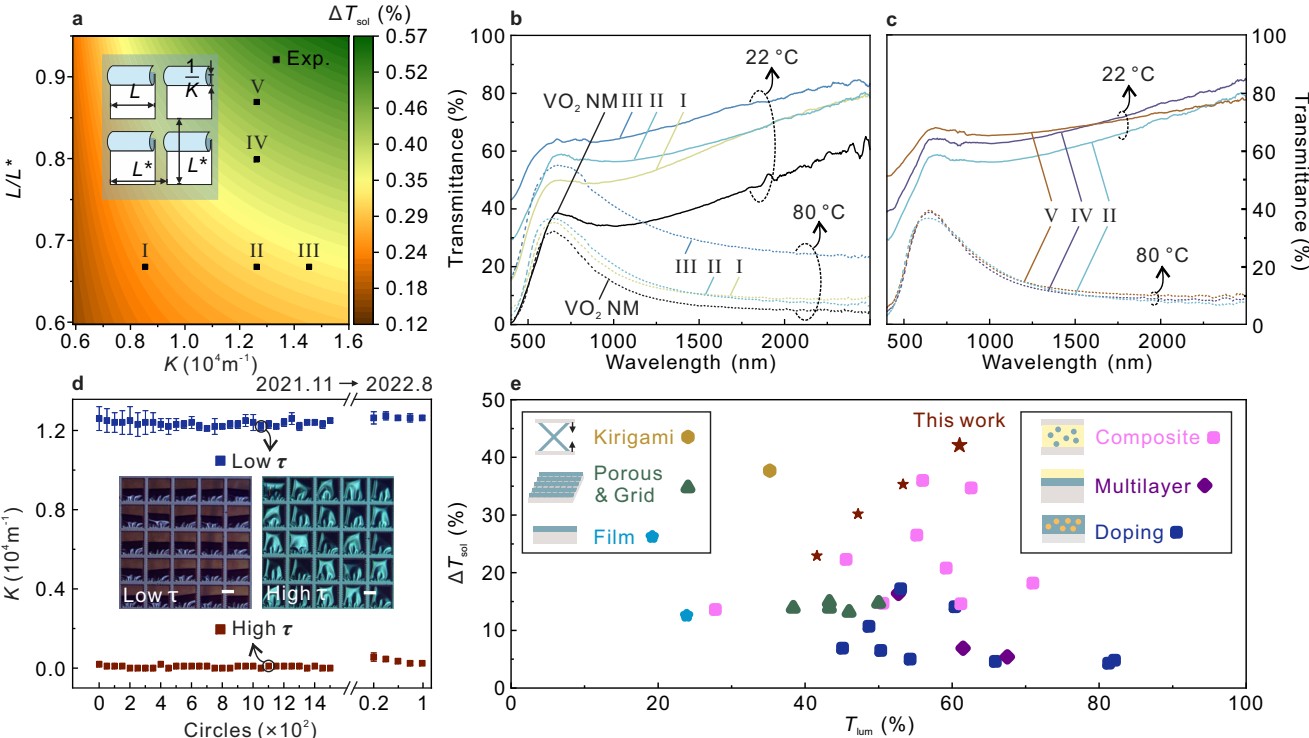

**Fig. 4 | Optimization of rolled-up SW for enhancing $T_{lum}$ and $\Delta T_{sol}$. a** Simulated $\triangle T_{sol}$ of rolled-up SW as functions of $K$ and $L/L^*$. The solid points represent the parameters selected of rolled-up SW in the experiment. The inset is schematic of rolled-up structures. The length of rolled-up structures, $L = 200$ μm. **b** Transmittance spectra of the rolled-up SW with the $K$ of $1.45 \times 10^4$ m$^{-1}$ (I), $1.26 \times 10^4$ m$^{-1}$ (II), $0.85 \times 10^4$ m$^{-1}$ (III), and 0 (VO$_2$ NM) at different temperatures. The $L/L^*$ is kept constant at 0.67. **c** Transmittance spectra of the rolled-up SW with $L/L^*$ of 0.87 (V), 0.80 (IV), and 0.67 (II) at different temperatures. The $K$ is kept constant at $1.26 \times 10^4$ m$^{-1}$. **d** Fatigue tests of the rolled-up SWs switching between low $\tau$ and high $\tau$. Insets are the microscope images of rolled-up SWs at low $\tau$ and high $\tau$. Scale bar: 100 μm. The corresponding error bar represents standard deviation. 25 rolled-up structures on sample IV after every 50 heating-cooling circles were measured once and the results were used to calculate the standard deviation. **e** Comparison of this work with the best-reported experimental works regarding the $T_{lum}$ and $\triangle T_{sol}$, and the schematics of corresponding different types of VO$_2$-based SW are displayed[5,13,17–19,38–59]. Source data are provided as a Source Data file.

exposed to light at low $\tau$. In addition to that, $\triangle K$ of sample iii, iv and v are equal to their initial $K$, which means that these 3 samples can change from rolled-up status into flat status with temperature stimuli. $\triangle K$ of sample i and ii are smaller than those of the other samples due to their low crystallinity and impure ingredients caused by depositing at low growth temperatures (Supplementary Fig. 16). In order to analyze the shape deformation process, sample iii with maximum $\triangle K$ was selected and its structure changes were recorded at different temperatures (Fig. 3c and Supplementary Movie 2). Here, to build a simple structure model, we approximate the cross section of the rolled-up structure as a standard circular arc. The side of the rolled-up structure connected to the film is set as the center of the circle, and the distance to the edge of the rolled-up structure is the radius (i.e., the distance between the dashed lines in Fig. 3c). Since radius ($R$) is not available for the flat film structure, we choose $K$ instead of the $R$ as the characterization parameter: $K$ of flat structure is set to zero and $\triangle K$ is the difference of $K$ in two shape statuses. In Fig. 3c, $K$ changes with temperature raise, and the structure demonstrates significantly shape deformation at ~45 °C, leading to decrease of exposed quartz substrate area. The phase transition completes when the temperature is increased to 48 °C, and the rolled-up structure fully expanded and covered the quartz substrate. On the other hand, when the temperature is reduced to 34 °C, the rolled-up structure returned to its initial geometry. The response time of the actuation is ultrafast at $\tau_c$: a rolled-up structure can complete the deformation in 6.7 ms stimulated by laser heating (Supplementary Fig. 17). For overall heating, the time required for deformation is closely related to the variation in temperature (Supplementary Fig. 18). In addition, it is found that continuous heating of tens of degrees did not lead to observable structural

change after the phase transition, indicating that the deformation of the rolled-up structure was almost independent of thermal expansion (Supplementary Fig. 19). The $\tau_c$ of the rolled-up SW reduction considers to be due to the released strain of VO$_2$ NM in rolling geometry[37]. Other experimental result that confirms $\tau_c$ of sample iii is in situ Raman spectra measurement, and $A_g$ vibration peak belonging to the VO$_2$(M) disappears at 50 °C and reappears during cooling process at <40 °C (Supplementary Fig. 20).

Raman spectroscopy can also accurately identify the strain status. Here, Raman spectra of rolled-up SW samples were obtained with the excitations of 532 and 633 nm lasers for different penetration depths (Fig. 3d), which allowed the analysis of strain at different locations of the NM. For a rolled-up SW sample, the two low-frequency phonons $\omega_1$ and $\omega_2$ correspond to V–V lattice motion, and the peak positions shift with different excitation lasers. The shifts of Raman peaks ($\triangle\omega$) excited by different lasers are summarized in Supplementary Fig. 21 and both $\triangle\omega_1$ and $\triangle\omega_2$ reduce with increased growth temperature. According to these shifts, the initial strain gradient ($\triangle\varepsilon$) along the radial direction are calculated (Supplementary Note 3), and the results are identified by black points in Fig. 3e. To quantitatively describe the variation of $K$ as a function of $\triangle\varepsilon$, we refer to the linear strain theory model used for bilayer systems. The solid lines in Fig. 3e are the calculated $K$ with different VO$_2$ NM thicknesses (Supplementary Note 3), and the model agrees well with the experimental results of 100 nm VO$_2$ NM. Moreover, it is observed that the $K$ increases with $\triangle\varepsilon$, and $\triangle\varepsilon$ of the sample i, ii, iii, iv, and v are determined to be 0.891, 0.666, 0.582, 0.388, and 0.309%, respectively. It is worth noting that for sample iii, $\Delta\varepsilon$ calculated by Raman spectra is close to the result from small-angle synchrotron radiation XRD (0.5448%), while experimental $\tau_c$ and $\triangle\varepsilon$ also match in

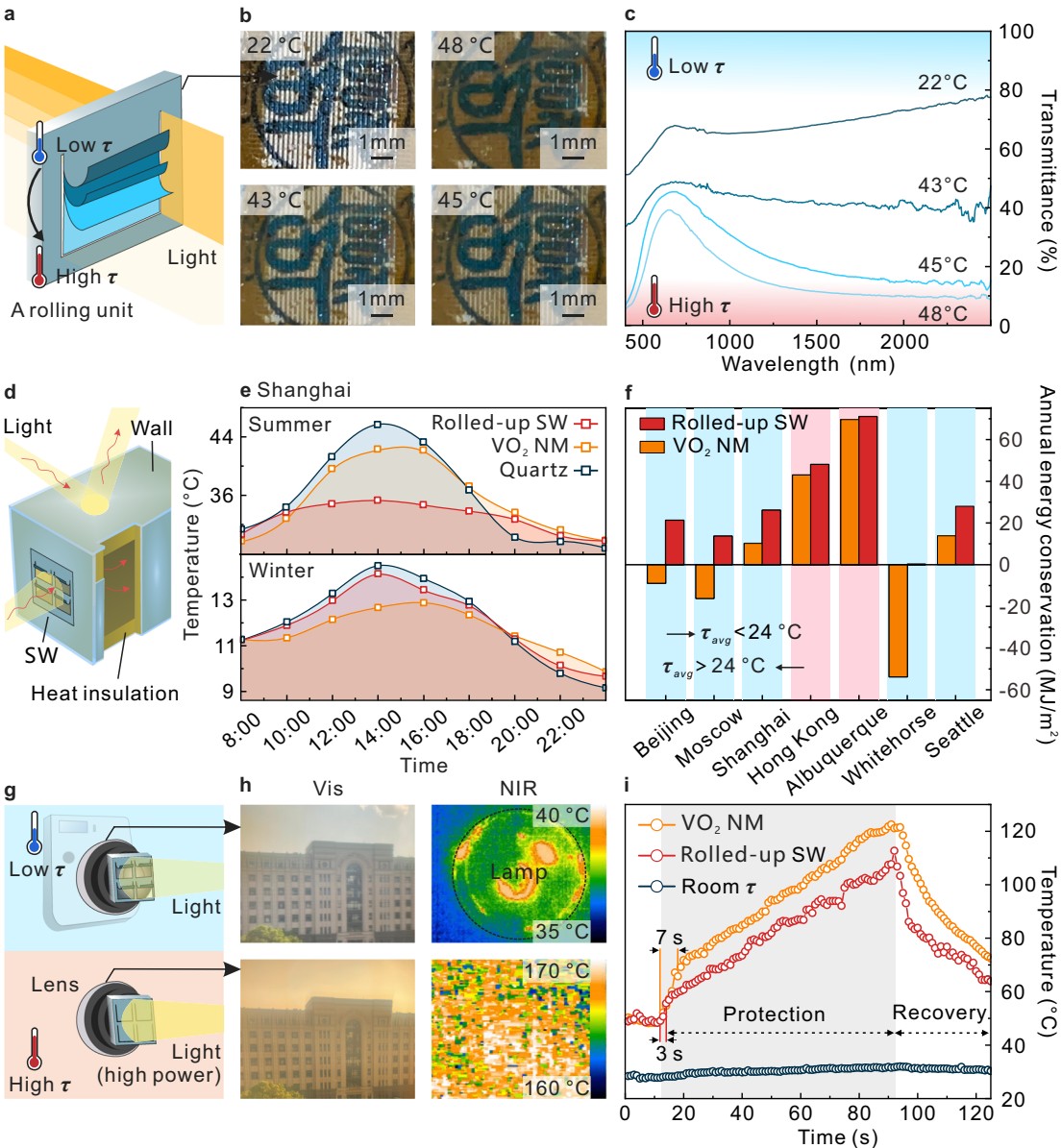

**Fig. 5 | Rolled-up SW for energy saving and lens protection. a** Schematic of multi-level light control by different temperatures. **b** Photograph of rolled-up SW at different temperatures. **c** Transmittance spectra of the rolled-up SW at different temperatures. **d** Schematic of the house model for the rolled-up VO$_2$ SW demo. **e** The daily indoor temperature modulation of different seasons in Shanghai by using different windows. **f** Annual energy conservation with different windows compared to the normal glass window in 7 cities. The blue areas indicate that the annual average temperature ($\tau_{avg}$) of the city is lower than 24 °C, and the red areas are the opposite. **g** Schematic of rolled-up SW as a smart lens protective cover. **h** Photograph and infrared images were taken by the rolled-up SW-installed camera at different temperatures. **i** Temperature dependence of VO$_2$-based smart lens protective cover on the time. The gray region is heated by high power infrared radiation. Source data are provided as a Source Data file.

phase diagrams of VO$_2$ (Supplementary Fig. 22). This proves the Raman spectra as a reliable method for calculating $\triangle\varepsilon$. Interestingly, according to the $\varepsilon_{MIT}$ (0.61%), the Fig. 3e can be divided into two regions, the $\triangle\varepsilon$ of samples iii, iv, and v are less than the $\varepsilon_{MIT}$ and those of samples i and ii are larger. This result indicates the necessary condition of a VO$_2$ rolled-up structure realizing the shape deformation from rolled-up to unrolled (flat) during phase transition is that $\triangle\varepsilon$ should be less than or equal to $\varepsilon_{MIT}$. It provides a theoretical basis for intelligent light control by the shape deformation of rolled-up SW.

### Optimization of rolled-up SW for enhancing $T_{lum}$ and $\triangle T_{sol}$

For the rolled-up SW, $K$ and the arrangement periodicity of rolled-up structures ($L/L^*$) are the key factors for $\triangle T_{sol}$ (Supplementary Note 4 for calculation method). Each structure of the rolled-up SW is

uniformly arranged on the quartz glass, and the VO$_2$ NMs are rectangles with the same side length of 200 μm ($L$), each with a lateral and longitudinal spacing of $L^*$. A simple model is built to simulate $\triangle T_{sol}$ as a function of $K$ and $L/L^*$ (Fig. 4a), and the simulation method is described in Supplementary Note 5. Simulation results show the $\triangle T_{sol}$ is proportional to $L/L^*$ and $K$. Moreover, highly controllable $K$ and $L^*$ can modulate $\triangle T_{sol}$ from 0.12 (blue region) to 0.57 (red region). In order to verify the accuracy of the simulation, we chose different parameters of rolled-up SWs to carry out transmittance measurement for calculating $\triangle T_{sol}$ (sample I, II, III, IV, and V). In the experiment, we study the effect of $K$ on light control with constant $L/L^*$(0.67), the transmittance of visible and NIR at low $\tau$ is greatly improved with the $K$ of the rolled-up SW increases (Fig. 4b). With the temperature rises, rolled-up SW with $K$ less than $1.26 \times 10^4$ m$^{-1}$ (sample II and III) can achieve enough

deformation and complete thermochromism for blocking the light, similar to the light control effect of VO$_2$ NM at high $\tau$. It is worth mentioning that the light transmittance of the rolled-up SW in flat status is not equal to that of the VO$_2$ NM because the corrosion window (~2.5% of the total area) leads to uncovered areas and effective light control by phase transition cannot realize there. However, since the rolled-up SW with $K$ of $1.45 \times 10^4\,m^{-1}$ (sample I) is difficult to fully shift to a flat status, it transmits most light at high $\tau$ leading to reduces $\triangle T_{sol}$. Therefore, the rolled-up SW with $K$ of $1.26 \times 10^4\,m^{-1}$ is selected to explore the influence of $L/L^*$. By increasing $L/L^*$, the light transmittance is improved at low $\tau$ (Fig. 4c). Meanwhile, these rolled-up SWs all can achieve a light-blocking effect close to flat VO$_2$ NM at high $\tau$. Photographs of the rolled-up SW with different parameters in close proximity to and away from the observed object also perfectly match the results of the transmittance tests (Supplementary Figs. 23, 24). Thus, under sufficient shape deformation of the rolled-up SWs, changing $L/L^*$ and $K$ are the simplest and the most effective way to adjust $\triangle T_{sol}$. $\triangle T_{sol}$ and $T_{lum}$ of different samples are listed in Supplementary Table 3. Here, the errors between the simulation and the experiments are due to the simplified model which only considers the solar modulation caused by the change of exposed areas corresponding to rolled-up and unrolled statuses. Furthermore, the rolled-up SW is stable during switch between the two statuses, and few failures occur in 1600 heating-cooling circles, 9 months interval (Fig. 4d). The rolled-up SW after cycle test and prolonged storage also has efficient light modulation ability (Supplementary Fig. 25). Moreover, in order to better meet the challenges in actual environments, rolled-up SW can be encapsulated by polydimethylsiloxane between two layers of glass (Supplementary Fig. 26), and the encapsulated rolled-up SW thus can withstand wind/rain and chemical attacks, and most importantly, maintain efficient light modulation ability by phase transition (Supplementary Fig. 27). The shape deformation stability and environmental adaptability of the rolled-up SW provides the potential for building windows in practical environments.

The rolled-up SW takes significant advantages in VO$_2$-based SWs for solar modulation based on the synergy of shape deformation and light transmission. The comparison of the rolled-up SW with other VO$_2$-based SWs in terms of $\triangle T_{sol}$ and $T_{lum}$ is shown in Fig. 4e [5,13,17–19,38–59]. Other parameters of performance that prove the superiority of the rolled-up SW are shown in Supplementary Fig. 28, and the specific data of various VO$_2$-based SWs is listed in Supplementary Table 4. Here, the schematics of other VO$_2$-based SWs includes VO$_2$ film, porous & grid, kirigami SW, composite, multifunctional coatings, and doping are shown in the Fig. 4e. The comparison displays that the rolled-up SW achieved a great breakthrough in $\triangle T_{sol}$, reaching the maximum value among all VO$_2$-based SWs. Meanwhile, $T_{lum}$ of the rolled-up SW also greatly meets the lighting requirements of civil windows. Moreover, compared with the conventional VO$_2$ NMs, $\triangle T_{sol}$ of the rolled-up SW is increased by 238%, and $T_{lum}$ enhances 155%. Such an efficient solar modulation ability benefits from the significantly enhanced light transmittance at low $\tau$. More, the comparison of $T_{lum}$ and $\triangle T_{sol}$ of this work to different materials-based SW is shown in Supplementary Fig. 29a, b. Hydrogels, ionic liquids and phase change polymers-based SW have achieved very high $\triangle T_{sol}$, but their $T_{lum}$ at high $\tau$ is approaching 0, suggesting they are loss of light capacity. Interestingly, many of high $\triangle T_{sol}$ SWs are not single material systems, they combine high visible light transmission materials with VO$_2$, expecting better performance through dual modulation in the visible and infrared regions, such as hydrogel/VO$_2$ and ionic liquid/VO$_2$. Our rolled-up technology with material compatibility can be combined with different material film systems to achieve further improvements on the performance of SWs (Supplementary Fig. 29c–f). Therefore, the current approach can further break through the performance limitations imposed by flat structures.

## Rolled-up SW for energy saving and lens protection

In addition to breakthrough in $\triangle T_{sol}$ and $T_{lum}$, the light transmittance changing with $K$ also provides the possibility of intelligent multi-level light control (Fig. 5a). From the previous study, the $K$ of the rolled-up SW adapts to ambient temperature and changes with season, weather, and time. At different stable temperatures, the badge placed below the rolled-up SW shows significant differences in clarity, which is due to the different $K$ of the rolled-up SW. (Fig. 5b). This result causes the light transmittance to change step by step with increasing temperature (Fig. 5c). In order to verify the actual temperature control of the SW-installed house, a small house that only exchanges heat through a window was designed (Fig. 5d). Besides the experimental results in a simulated environment (Supplementary Fig. 30), the performances of different windows were evaluated in a real environment. In Shanghai, the ambient temperature varies greatly in different seasons. The surface temperature of building is close to 50 °C in summer and close to 0 °C in winter. For better sensing of ambient temperature changes, the small house with various windows was placed on the windowsill of the building to receive direct sunlight and the indoor temperature was monitored during sunny days in summer and winter (Fig. 5e). When the temperature rose to the highest value in the summer noon, direct sunlight radiation usually led the surface temperature of the house to ~45 to 50 °C. The temperature of rolled-up SW-installed house before 12:00 pm was between the other two houses, which is similar to the situation in winter. After 12:00 pm, significant thermal insulation effect of the rolled-up SW beneficial from shape deformation because the phase transition occurred. Compared with the quartz-installed house, the maximum temperature difference was up to 10.6 °C. In the winter, ambient temperature changes from 9 to 15 °C, the quartz-installed house with highest solar transmittance had the greatest range of temperature variations. Due to the ambient temperature lower than the $\tau_c$, the rolled-up SW stayed rolled-up status all day, and its transmittance was lower than quartz glass but much higher than VO$_2$ NM. Therefore, the indoor temperature variation trend in house with rolled-up SW was close to that of house with quartz glass, while VO$_2$ NM-installed houses had the lowest indoor temperature variation in winter. It indicates that the rolled-up SW enhances light transmission and heat exchange at low $\tau$, and blocks sunlight for heat insulation at high $\tau$. For further research the energy balance of house installed with SW, a building model was built and EnergyPlus simulation was carried out (the building model and simulation parameters are listed in Supplementary Tables 5 and 6). The obtained annual energy consumptions of houses installed with SWs and normal glass in different cities are shown in Supplementary Fig. 31, and the calculated saved energies are shown in Fig. 5f, where the blue areas indicate that the annual average temperature ($\tau_{avg}$) of the city is lower than 24 °C, and the red areas are the opposite. The simulation results visually show that the energy conservation of VO$_2$-based SWs is more efficient in locations with high average annual temperatures. Comparing the energy conservation of rolled-up SW with VO$_2$ NM SW, we notice that the rolled-up SW can save energy in any cities with different climates, while VO$_2$ NM SW is not suitable for cities with more low-temperature days. Moreover, the energy saving rate of rolled-up SW is also at a high level compared to SWs based on different materials with high $\triangle T_{sol}$ (Supplementary Table 7), indicating the universality of rolled-up SWs for future house windows. In addition, for practical applications, industrial-level scalability is crucial. In our present work, the size of sample used for demo is ~1 cm$^2$, but we should stress that the fabrication process with photolithography step can prepare patterned VO$_2$ NMs of larger area (Supplementary Fig. 32). To produce rolled-up SWs with even larger size (e.g., ~m$^2$ scale in practice), industrial grade magnetron sputtering and inexpensive soft lithography technique, which allows nanoscopic precision over indefinitely large areas [60,61], is promising, but more research inputs are still needed for industrial fabrication.

Rolled-up SW enhances transmittance changes during phase transition while also improving reflectance changes (Supplementary Fig. 33). This characteristic also provides possibility for the rolled-up SW as a protective cover for all kinds of lenses (Fig. 5g). At low $\tau$, the rolled-up SW-installed lens with high transmittance of visible and NIR hardly affects the imaging of a camera. However, when the temperature rises by high power light radiation, especially NIR radiation, the rolled-up SW cover automatically unrolls and reflects the light to protect the lens and camera. In Fig. 5h, the photos taken by mobile phone camera (visible) and NIR detector where rolled-up SW were installed on the lenses. The upper photos were taken at low $\tau$, and lower photos were taken after phase transition. In both visible and NIR ranges, there is a good imaging effect at low $\tau$ and the enhanced reflection at high $\tau$ plays a protective effect (Supplementary Movies 3 and 4). In order to further explore the influence of temperature change on the lens, rolled-up SW and VO$_2$ NM were affixed to the camera lens, and a high-power infrared lamp was used to irradiate the lens, while another infrared camera was used to record the temperature change of the cover with time (Fig. 5i). With the accumulation of time, the radiation brought a significant temperature increase on the surface of the cover. In Fig. 5i, the gray region is the heating period which can be divided into two stages. The rapid temperature rising stage is the warning time, and the infrared imaging camera can shoot normally. Another stage is the protection region, during which the temperature rise slows down due to enhanced NIR reflectivity. The warning time of the rolled-up SW and VO$_2$ NM is 3 and 7 s, respectively. The faster response time of rolled-up SW benefits from the low $\tau_c$ and high actuation sensitivity. It also takes a relatively long time for the rolled-up SW to recover the rolling structure in cooling and reach the working status again. Based on these results, the rolled-up SW can efficiently block the high-power light to avoid lens damage and have little influence on the imaging of camera. In addition, the rolled-up SW can change the absorption to increase the modulation range of emissivity during phase transition (Supplementary Fig. 34) and achieve infrared imaging by patterned laser radiation (Supplementary Fig. 35).

## Discussion

In summary, rolled-up SW combined mechanochromic with thermochromic properties was developed. By using the rolled-up technology, the VO$_2$ microstructure arrays with controllable initial $K$ and deformation upon phase transition were fabricated on quartz. As a SW, it appears rolled-up status at low $\tau$ for enhanced transmission of visible light and NIR. At high $\tau$, the rolled-up SW automatically unroll to shield visible light and NIR. Through further optimization with $K$ and $L/L^*$, the SW with high $\triangle T_{sol}$ (42.14%) and $T_{lum}$ (61.01%) is fabricated with a $K$ of $1.26 \times 10^4$ m$^{-1}$ and $L/L^*$ of 0.87. Due to the internal strain introduction into VO$_2$ NM, $\tau_c$ is decreased to under 50 °C. Based on the low $\tau_c$ and self-adapting multi-level solar modulation, the temperature of rolled-up SW-installed house is lowered by 10.1 °C compared with a quartz-installed house in a sunny day of summer of Shanghai, and they have similar in-door temperatures in winter. In addition, rolled-up SW can be used as a smart lens protective cover, which responds quickly and reflects high power radiation when the phase transition occurs, but has few effects to imaging at low $\tau$. These findings indicate that the rolled-up SW can act as an intelligent shutter with multiple functions for smart building and intelligent sensor and actuator with the perspective of energy efficiency.

## Methods

### Growth of VO$_2$ NMs and characterization

Firstly, quartz substrate was cleaned by ultrasonication in acetone (99.5%), ethanol (99.7%), and deionized water for 5 min, respectively. After cleaning, VO$_2$ NMs were grown on quartz substrates by direct-current magnetron sputtering (Kurt J. Lesker, PVD75) of a pure V metal (99.9%) target in oxygen-argon flux. VO$_2$ NMs were deposited at different temperatures of substrates from 350 to 550 °C for 1200 s (Heating rate is 10 °C/min). The oxygen-argon ratio and sputtering power were kept at 40 sccm: 60 sccm and 200 W (DC), respectively. The structure of rolled-up SW and surface morphologies of VO$_2$ NMs were characterized via SEM (JEOL JSM-6701F). Specially, the rolled-up state is taken at low voltage (5 kV), and the unrolled state is taken at high voltage (15 kV). The VO$_2$ NMs were also characterized by small-angle synchrotron radiation XRD (Shanghai Synchrotron Radiation Facility). The angle of incidence was fixed at 0°, 18°, 32°, and 45°, and the measurement was taken by using a $2\theta$ scan configuration. The electrical properties of NMs were measured on a Keithley 4200 semi-conductor characterization system, and the samples were heated with a rate of 5 °C/min. Before the characterization by TEM, the sample lamella was cut by using focused ion beam (FIB, FEI Helios NanoLab 600), and deposition of Pt layer was used to protect the surface of the VO$_2$ NM during preparation process. The TEM and STEM images were collected by using the JEOL JEM-ARM300F at 300 kV. The sample holder for in-situ heating was DENS, lighting d9, and the heating rate is -10 °C/min.

### Fabrication and characterization of rolled-up structures

As the first step of self-rolling, a layer of photoresist (AZ-5214, Microchemicals GmbH, Germany) with about 1 μm thickness was spin-coated (4000 RPM, 30 s, KW-4A, SIYOUYEN, China) and etching windows were defined by photolithography (HEIDELBERG, UPG501). After development, the window was exposed and etched by reactive ion etching (RIE, Trion T2) for 60 s (30 sccm CF$_4$ and 30 sccm Ar flow, 300 mTorr chamber pressure, and 100 W etching power). The photoresist layer was removed by ultrasonication in ethanol (99.7%) lasting for 30 s. For the rolling process, patterned VO$_2$ NMs were released from the substrate by using 40% HF (hydrofluoric acid) solution at room temperature for 5 min. Due to the high selectivity, the property of the VO$_2$ NMs can hardly be influenced by this etching process. Finally, critical point drying (Leica CPD030 Critical Point Dryer) was applied to dry the rolled-up structures, avoiding structural collapse.

### Characterization of rolled-up SW

The morphology of rolled-up SW was characterized by SEM (SEM, JEOL JSM-6701F). The phase transition temperature and initial strain of rolled-up SW were identified by Raman scattering spectroscopy (Horiba JY HR-800) with heating/cooling stage (Linkam LTSE420) and light source of 532 and 633 nm laser. The actuation ability of the rolled-up structures at different temperatures was recorded by an optical microscopy (OLYMPUS, BX51) with CCD camera (TOUPCAM, UHCCD05100KPA). For transmittance measurement, an integrating sphere was used to measure the total transmittance in the wavelength range from 350 to 2500 nm at normal incidence by using a Hitachi U-4100 spectrometer. The infrared imaging was taken by the Infrared Camera (InfRecR300SR). The reflectance and absorption of rolled-up SW in the near infrared (2.5–17 μm) was measured by Fourier transform infrared spectroscopy (Thermo Scientific, NICOLET iS10).

## Data availability

All the data supporting the findings of this study are provided in the Supplementary Information and Source Data file. Source data are provided with this paper.

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

## Acknowledgements

This work is supported by the National Key Technologies R&D Program of China (2021YFE0191800, Y.M. and 2021YFA0715302, G.H.), the National Natural Science Foundation of China (61975035, Y.M., 51961145108, Y.M., 61871134, C.Z. and 51925208, Z.D.), and Science and Technology Commission of Shanghai Municipality (21142200200, Y.M., 20501130700, Z.D. and 22ZR1405000, G.H.). X.C. acknowledges the Youth Innovation Promotion Association, Chinese Academy of Sciences (2018288, X.C.). Part of the experimental work was carried out in Fudan Nanofabrication Laboratory, Shanghai Institute of Microsystem and Information Technology, CAS, Shanghai Synchrotron Radiation Facility, and Shanghai Institute of Ceramics, CAS.

## Author contributions

Y.M. and X.C. conceived the project and designed experiments. X.L. fabricated and characterized the rolled-up smart window samples with assistance from C.L. and Y.W. W.H. and C.Z. performed the TEM experiments and analyzed the data and K.W. performed the Raman experiments. C.C. characterized and simulated the performance of rolled-up smart window. G.H., X.C., and Y.M. co-wrote the paper. B.X., Z.T., E.S., J.C., and Z.D. discussed the results and commented on the manuscript.

## Competing interests

The authors declare no competing interests.
