## [Peer Review File · Nature Communications]

Self-rolling of vanadium dioxide nanomembranes for enhanced multi-level solar modulationREVIEWER COMMENTS

Reviewer #1 (Remarks to the Author):

This manuscript reported a rolled-up VO₂-based smart window with thermochromic properties. Compared with reported studies, a certain degree of advances in VO₂-based smart window was achieved. However, this advance may not be amazing enough to impact the study on smart windows (a broader application field). Meanwhile, a deeper discussion is necessary for readers to better understand this work. I think this manuscript is hardly up to the high standard of "Nature Communications". A partial list of the suggestions is shown as follows.

1. How much is the bonding strength between VO₂ NM and the quartz? I am worried that VO₂ NM is easily stripped.
2. After HF solution etching, why only VO₂ NM in active zone is released from the quartz, while the so-called framework of VO₂ NM still adheres on the quartz.
3. From the results, the thermal actuator is made of only a layer of VO₂ NM with thickness of a few hundred nanometers. Is it brittle? In other words, is it strong enough to prevent environmental impacts (such as stretching and wind) from damaging its integrity?
4. What is the role of Pt in Fig. 2b? It is referred to only once more in all text including manuscript and supplementary information.
5. From the discussion of Fig. 2 and Fig. 3, I conclude that initial strain gradient ($\Delta\epsilon$) along the radial is the driving force behind the formation of rolled-up VO₂ NM at low temperature. Which process is this strain gradient introduced in? And how it was introduced.
6. Further, the authors analyzed the strain status at different depths in VO₂ NM by Raman spectroscopy with different excitations of 532 nm and 633 nm lasers (Page 9, Fig. 3d-e). On Page 5, Paragraph 1, the authors wrote that "When the temperature is above trigger temperature (τ_c), the rolled-up VO₂ NMs automatically unroll to flat status by strain change in the phase transition and completely cover the areas which is exposed in previous rolled-up status (Fig. 1d)." Does the strain gradient disappear or reversibly evolve into another steady state, when the ambient temperature is higher than trigger temperature (τ_c)? Correspondingly, how is the measuring Raman spectroscopy with different excitations in this case?
7. Supplementary Fig. 8b shows that a part of VO₂ NM with growth temperature of 400 °C becomes flat, when the ambient temperature is 80 °C. Whether VO₂ NM with lower growth temperature (350 °C and 400 °C) could be completely flat, when the ambient temperature is higher than 80 °C.
8. On the other hand, whether VO₂ NM with higher growth temperature (450 °C, 500 °C and 550 °C) bends in the opposite direction, when the ambient temperature is higher than 80 °C.
9. On page 12, paragraph 1, the authors wrote that "Moreover, highly controllable K and L* can modulate ΔT_{sol} from 0.12 (blue region) to 0.57 (red region)". "blue region" may be the wrong spelling.
10. The feature comparison of T_{lum} and ΔT_{sol} between different types of VO₂-based smart window in Fig. 4e is meaningful. But, in my opinion, another more comprehensive feature comparison of T_{lum} and ΔT_{sol} between various smart windows composed of different materials (such as hydrogel, liquid crystal, phase-changing polymer and perovskite) is more important. This additional comparison, showing the advances in a broader field, is helpful to estimate the significance of this work. In addition, what is the potential of VO₂ in smart window, taking T_{lum} and ΔT_{sol} into account?

Reviewer #2 (Remarks to the Author):

General assessment:

The manuscript is a valuable contribution to the area of micro-mechanic actuators applied to the energy field, as it presents a quite curious new system composed of arrays of self-actuating micro-curtains with outstanding thermochromic performance for smart windows, but with interest in other applications (protective covers for lenses, etc) as proposed by the authors. The high level of this study and results presented herein, as well as the overall text quality, justify

the acceptance of this work in a high-impact journal such as the present one, given that the issues indicated below are properly addressed.

Comments:

1) My main concern lies in the scalability of this technology to address the intended application. For practical building integration it would be necessary to pattern several m² of these structures in a timely and inexpensive fashion, which is not at all possible with the photolithography approach employed here. However, there is presently a wide range of soft-lithography methods that should be considered in view of allowing large-area long-range micro-patterning, such as those reported in related works (e.g. doi.org/10.3390/nano11071665 and doi.org/10.1002/admi.202000264). In view of this, the future adaptability of this solution to allow industrial-level scalability is a crucial aspect that needs to be discussed in the manuscript.

2) Another critical aspect which is insufficiently addressed is concerned with the stability of the micro-structured membranes upon repeated cycles of rolling and unrolling. Namely, how does the thermo/mechano-chromic performance decrease with the number of rolling cycles and, from that, can the authors estimate the operational lifetime of this technology in typical conditions?

3) An energy balance study should also be presented, indicating the power savings that can be expected in average buildings from integrating Smart Windows with this technology, as compared to other SW solutions under development.

4) Lastly, it is not clear why the present structures are named "origami", since origami usually involve more complex shapes. I recommend re-naming the NMs to a term that is more specific to its actuation, such as "rolling scrolls" or "self-rolling micro-curtains" or something of that sort.

Reviewer #3 (Remarks to the Author):

In this paper, authors present interesting and unique combination of mechanochromism and thermochromism in the multifaceted vanadium dioxide. The manuscript has applied focus in smart windows and camera shield for self-regulated temperature control. This work provides detail information on the material, thermal, optical, and mechanical characterisations, and fabrication approach. The provided information is enough to reproduce the results. The work is of significant importance to the current hot topics in vanadium dioxide for smart windows and origami-kirigami methods.

While the evidences and conclusions are presented well, some more information on the following aspects would benefit the work. I would suggest a minor revision for this work.

- What is the size of the sample (not single flap) authors fabricated to show the proof-of-concept of this? As the focus of this work is applied, how do authors plan do translate this work for large windows? As sputtering has been used here, this deposition method is not very appropriate for large scale.
- The typical transition temperature for VO₂ is 68C. More information and relevant references should be provided in the paper to reason why authors are getting transition at lower temperature of 48C
- More information should be provided for Figure. 3b statistical data. Especially how many samples are tested to calculate the error graph for each temperature.
- Is this un-rolling effect is gradual or occurs suddenly at transition temperature? What is the rolling recover time? The images provided in the paper are only at 20C and 80C. Can authors provide images at intermittent temperatures as well? It would be interesting to see if the rolling is caused by thermal stress or it is triggered by the phase-change.
- Is there any relationship between the effect of K and L/L* modulation on critical transition temperature?
- Typo – Line 331 and 332 page 16 – Should it be 12 pm? And not 12 am.

Reviewer #4 (Remarks to the Author):

In the manuscript, the authors reported the vanadium dioxide (VO₂)-based three-dimensional (3D) smart windows (SWs) combining mechanochromism with thermochromism to enhance multi-level solar modulation. Due to the rolling origami of VO₂ nanomembranes integrating on quartz, the 3D SW was able to synergy of shape deformation and light transmission when the phase transition temperature was reached, which significantly increased the solar modulation and luminous transmittance. The manuscript also presented impressive experimental data and careful data analyses to extract the principle of control the deformation with temperature in a VO₂-based microstructure. This strategy shown in the paper provided a new idea for further energy efficiency improvements in SWs based on planar structures. Therefore, this manuscript should be recommended for publication in Nature Communications.

While I have some concerns and comments for improvements:

1. It was known that a crystal structural variation was vital in the VO₂ phase transition, but it was difficult to see the huge difference in the position and arrangement of the atoms in the TEM images. So how should the completion of the phase transition be judged in the in-situ TEM measurement?
2. In Fig. 3b, the authors mentioned that the relatively small deformation of the low temperature grown VO₂ nanomembranes during the phase transition was due to the poor nanomembranes quality and the impure composition. For this concern, my suggestion was to add some supporting tests to prove this point, such as XRD, Raman, etc.
3. The authors focused on improving the solar modulation by varying the radius and the structures alignment period (L/L^*). The L/L^* could theoretically be further increased to 1 for a better solar modulation. Did the authors attempt to further increase L/L^* ?
4. I suggested that the authors could describe the simulation process in more detail and provide n , k values for the VO₂ nanomembrane in low and high temperatures to improve the credibility of the simulation results.

-----Responses to the reviewers' comments-----

Reviewer #1

This manuscript reported a rolled-up VO₂-based smart window with thermochromic properties. Compared with reported studies, a certain degree of advances in VO₂-based smart window was achieved. However, this advance may not be amazing enough to impact the study on smart windows (a broader application field). Meanwhile, a deeper discussion is necessary for readers to better understand this work. I think this manuscript is hardly up to the high standard of "Nature Communications". A partial list of the suggestions is shown as follows.

Our response:

Thank you very much for your comments and suggestions on our work, and we totally understand the reviewer's concern. With rapid progress of scientific research, single material and simple structure are no longer sufficient to satisfy the increasing demands in a variety of fields, and this is also the case in smart windows (SWs). Therein, researchers have focused on adopting multiple materials to achieve photomodulation in a wide range of wavelengths, and hydrogel/VO₂, chalcogenide/VO₂, radiative cooling layers/VO₂ etc. have been studied. In addition, the combination of VO₂ particles with shape-deformable polymers through kirigami assembly has opened up the study of quasi-3D SWs. As a 3D construction technique that has been proven to be compatible with different thin film systems, the rolled-up technique is expected to fabricate 3D SWs for performance enhancement. The self-rolled VO₂ nanomembranes demonstrated in this paper have indeed realised this vision. However, we should admit that there are still shortcomings in our work. Through careful revision in response to your comments, we hope to enrich our work and meet your expectations.

Comment:

1. How much is the bonding strength between VO₂ NM and the quartz? I am worried that VO₂ NM is easily stripped.

Our response:

We understand the reviewer's concern. It is difficult to measure the bonding strength of a nanomembrane (NM) to a substrate directly, so we usually use the nano-scratch method (i.e., using a 10 μm-diameter needle to cut through a film and measuring the force) to determine the work of adhesion between the film and substrate as a concise representation of the bonding effect. The nano-scratch tests of attached region of etched NM (Re-Fig. 1a) and un-etched NM (Re-Fig. 1b) show that the NMs in both cases have similar bonding strengths, suggesting that the rolled-up SW still has the same adhesion strength as the original VO₂ NM. In addition, the work of adhesion $W_{A,P}$ can be derived from the following equation:

$$P_{cr} = \frac{\pi r^2}{2} \left(\frac{2EW_{A,P}}{h} \right)^{1/2}$$

where r is the contact radius, h is the film thickness, E is Young's modulus, and P_{cr} is minimum critical load. Here, in our experiments, the $W_{A,P}$ of attached region of etched NM is calculated to be 0.29 and $W_{A,P}$ of un-etched NM is 0.31. The values are comparable to the work of adhesion of many thin film interfaces in practical devices (DOI: 10.1016/S1359-6454(01)00354-8). We consider that the adhesion of VO₂ NM to the substrate is adequate for practical application.

Re-Figure 1. a Nano-scratch test on attached region of etched VO₂ NM. **b** Nano-scratch test on VO₂ NM without etching.

Comment:

2. After HF solution etching, why only VO₂ NM in active zone is released from the quartz, while the so-called framework of VO₂ NM still adheres on the quartz.

Our response:

Thanks for your comment. As a 3D microstructure fabrication technique, the self-rolling technique achieves 3D tubular structure by selectively etching the sacrificial layer to release the above NM with initial stress. However, in HF etching step, the quartz in the framework regions will also be gradually etched and the VO₂ NMs in the framework regions are also partially release from the substrate. It is worth noting that the framework NMs are interconnected and therefore rolling is unlikely to be realized in these framework regions with short etching time. In order to further fix NMs in these regions, in the fabrication process of our experiment, we perform an additional step of lithography before wet etching to cover the framework with a thick photoresist layer, which helps to maintain the framework geometry (Re-Fig. 2a). The experimental results in Re-Fig. 2b confirm the effectiveness of the method, especially for short etching time. It can be seen that the framework is intact until the etching time reaches 1800 s, which is much longer than normal etching time of ~300 s.

Re-Figure 2. a The schematic of fabrication process by using rolled-up nanotechnology.

b Microscope images of sample with different etching times.

Our modification:

1. Re-Fig. 2 has been added as supplementary Fig. 8.
2. Finally, HF solution was used to etch the quartz, which released the VO₂ NMs for rolling process. In order to fix the framework (i.e., un-rolled) regions, photoresist layer might be used to cover these regions which helps to maintain the intact framework geometry (supplementary Fig. 8).

Comment:

3. From the results, the thermal actuator is made of only a layer of VO₂ NM with thickness of a few hundred nanometers. Is it brittle? In other words, is it strong enough to prevent environmental impacts (such as starching and wind) from damaging its integrity?

Our response:

Thank you very much for your question, and it helps us a lot. The rolled-up SW can withstand winds of a certain speed and will be influenced by surface tension of liquid. Re-Fig. 3 shows rolled-up structure have different statuses in wind (the shape is distorted but the NM's color does not change) and different temperatures (both the color and structure change obviously), which demonstrates that the rolled-up structure has wind resistance when the wind speed is lower than 12 m/s (by hairdryer). The other problems that limit its practical application are its poor resistance to acids and alkaline and its susceptibility to oxidation. Therefore, we consider encapsulating of the rolled-up SWs to reduce the influence from external environment. Here, the encapsulating method is schematically shown in Re-Fig. 4a. We place the rolled-up SW sample in the center of normal glass, and the sample is surrounded by PDMS as a bracket to support the upper glass cover. The PDMS is then cured at high temperature to form a completely sealed cavity (Re-Fig. 4b), and thus the sample is insulated from water, wind, air, and other factors that can cause damage to the rolled-up structure. The encapsulated samples were tested for acid and alkali resistance and the results are shown in Re-Fig. 5. In Re-Fig. 5a, both the encapsulated and unencapsulated samples are placed in NaOH solution with pH of ~9, and it can be seen that the un-protected VO₂ NM reacts with the NaOH and its color faded after 4 h. The encapsulated rolled-up SW is able to remain

uncorroded in NaOH for long period of time. Similarly, in Re-Fig. 5b, both samples are placed into 85% phosphoric acid. It only took a few minutes to remove VO₂ NM, while the encapsulated rolled-up SW keeps unchanged even after 1 h. Re-Fig. 5d shows the cleaning process carried out after each test, and the encapsulated rolled-up SW sample can survive after the flush of water flow. In addition, after a series of above-mentioned tests, the mechano/thermochromism of this encapsulated rolled-up SW sample was tested at different temperatures, and as expected, a significant light transmission decrease at elevated temperature could be seen (Re-Fig. 5e). These results indicate that the encapsulated rolled-up SW should be able to meet the requirements of the complex situations in practical environment.

Re-Figure 3. a Rolled-up SW in a simulated wind environment (by hairdryer). **b** Rolled-up SW in a heating process.

Re-Figure 4. a Schematic of an encapsulated device. **b** Encapsulated rolled-up SW.

Re-Figure 5. Encapsulated rolled-up SW and VO₂ NM placed in (a) NaOH and (b) H₃PO₄ solutions. c Reference pH values. d The process of cleaning with flush of water flow between each test. e Light transmission modulation ability after acid and alkaline tests.

Our modification:

1. Re-Fig. 4 has been added as supplementary Fig. 26.
2. Re-Fig. 5 has been added as supplementary Fig. 27.
3. Moreover, in order to better meet the challenges in actual environments, rolled-up SW can be encapsulated by polydimethylsiloxane between two layers of glass (supplementary Fig. 26), and the encapsulated rolled-up SW thus can withstand wind/rain and chemical attacks, and most importantly, maintain efficient light modulation ability by phase transition (supplementary Fig. 27). The shape deformation stability and environmental adaptability of the rolled-up SW provides the potential for building windows in practical environments.

Comment:

4. What is the role of Pt in Fig. 2b? It is referred to only once more in all text including manuscript and supplementary information.

Our response:

We sincerely apologize that the description of this result was omitted from the manuscript. The Pt layer mainly serves to protect the surface of the VO₂ NM from damage caused by the Focused Ion Beam (FIB) used for preparation of cross-sectional samples (Re-Fig. 6). The first step in cross-sectional sample preparation is to select a cut area (approximately 5 μm × 10 μm). Afterwards, Pt is deposited onto this area, and then FIB is used to cut the sample along the perimeter of the area until this part can be completely separated. Due to the high energy of the ion beam, the cutting process produces a lot of splatters, including VO₂ as well as substrate material. Without the protection of the Pt layer, these impurities will accumulate on the surface of the VO₂ NM and the surface of the VO₂ NM is difficult to be accurately determined in TEM characterization. In addition, after separation, the sample needs to be further thinned from the upper surface by using ion beam. The presence of the Pt layer ensures that the surface VO₂ NM is not destroyed during the thinning process and the mechanical stability can be well preserved. Therefore, a layer of Pt is left on the VO₂ surface and can be observed in TEM characterization.

Re-Figure 6. Low-resolution TEM image of VO₂ NM.

Our modification:

1. In the transmission electron microscopy (TEM) image, the VO₂ NM (between Quartz (lower part) and Pt (upper part. Pt layer mainly serves to protect the surface of the VO₂ NM during preparation of the cross-sectional sample.)) possesses orientation in the vertical direction and a buffer layer is noticed near the substrate (Fig. 2b)

Comment:

5. From the discussion of Fig. 2 and Fig. 3, I conclude that initial strain gradient ($\Delta\varepsilon$) along the radial is the driving force behind the formation of rolled-up VO₂ NM at low temperature. Which process is this strain gradient introduced in? And how it was introduced.

Our response:

The mechanism for introducing strain is schematically demonstrated in Re-Fig. 7. Due to oxygen insufficiency during the initial growth stage, the buffer layer formed and was attributed to the formation of oxygen-deficient Magnéli phases V_nO_{2n-1} (DOI: 10.1063/1.3642980, pink part in Re-Fig. 7a). The difference in these components leads to a lattice mismatch between the upper pure VO₂ NM and buffer layer, which creates a stress gradient in the vertical direction (Re-Fig. 7b) (DOI: 10.1002/adma.200801589 and 10.1088/1361-6463/aba312). The two layers are in tensile status with different values, and the stress gradient therein can be experimentally probed by synchrotron radiation XRD (supplementary Fig. 5 in revised Supplementary Information). When the NM with the vertical stress gradient is released from the substrate, the larger contraction of the upper part of the NM compared to the lower part creates a net bending force with the direction away from the substrate. At the same time, for NMs with same thickness, different growth temperatures enable the preparation of VO₂ NM with different initial stress gradients due to the various thermal expansion coefficients, which leads to formation of rolled-up VO₂ NM with different curvatures.

Re-Figure 7. a TEM image of the cross section of a VO₂ NM. **b** Schematic representation of rolled-up processes of a VO₂ NM. The red arrows qualitatively represent contraction of the layers after the releasing process.

Our modification:

1. Re-Fig. 7 has been added as supplementary Fig. 3.
2. Due to oxygen insufficiency during the initial growth stage, the buffer layer formed and was attributed to the formation of oxygen-deficient Magnéli phases V_nO_{2n-1},³⁵ but the buffer layer hardly affects the NM components and crystalline quality. **The difference in components leads to a lattice mismatch between the upper pure VO₂ NM and buffer layer, which creates a stress gradient along the vertical direction for self-rolling (supplementary Fig. 3).**

Comment:

6. Further, the authors analyzed the strain status at different depths in VO₂ NM by Raman spectroscopy with different excitations of 532 nm and 633 nm lasers (Page 9, Fig. 3d-e). On Page 5, Paragraph 1, the authors wrote that “When the temperature is above trigger temperature (τ_c), the rolled-up VO₂ NMs automatically unroll to flat status by strain change in the phase transition and completely cover the areas which is exposed in previous rolled-up status (Fig. 1d).” Does the strain gradient disappear or reversibly evolve into another steady state, when the ambient temperature is higher than trigger temperature (τ_c)? Correspondingly, how is the measuring Raman spectroscopy with different excitations in this case?

Our response:

We appreciate your suggestion. The strain gradient does not disappear during the phase transition, and the deformation is caused by the strain from the phase transition offsetting the initial strain gradient. For in-situ heating or cooling Raman test excited by lasers with wavelengths of 532 nm (Re-Fig. 8a) and 633 nm (Re-Fig. 8b), Raman peaks of VO₂ disappear at 50 °C and appear again while the temperature reduces to 40 °C. The statistics of the positions of the V1 and V2 peaks as a function of temperature are shown in Re-Fig. 8c and 8d. It can be visualised that the positions of the peaks

fluctuate within the tolerance range ($\sim 2 \text{ cm}^{-1}$, which is the resolution of our spectrometer) before the phase transition, indicating a trivial strain change in this stage (tens of degrees). However, it is worth noting that characterization of the strain change during the phase transition by Raman is difficult because the metal phase of VO_2 demonstrates no Raman activity. Other evidence demonstrating that the strain gradient still exists can be found in our response to Comment 7 below, where sample ii does not fully unfold after phase transition, suggesting a residual strain gradient therein.

Re-Figure 8. a Raman spectra of a rolled-up structure measured in heating or cooling process excited by laser with wavelength of 532 nm. **b** Raman spectra of a rolled-up structure measured in heating or cooling process excited by laser with wavelength of 633 nm. **c** Raman peak positions (ω_{V1} and ω_{V2}) as a function of temperature (laser: 532 nm). **d** Raman peak positions (ω_{V1} and ω_{V2}) as a function of temperature (laser: 633 nm).

Our modification:

1. Re-Fig. 8 has been added as supplementary Fig. 20.
2. Other experimental result that confirms τ_c of sample iii is in-situ Raman spectra measurement, and A_g vibration peak belonging to the $VO_2(M)$ disappears at 50 °C and reappears during cooling process at < 40 °C (supplementary Fig. 20).

Comment:

7. Supplementary Fig. 8b shows that a part of VO_2 NM with growth temperature of 400 °C becomes flat, when the ambient temperature is 80 °C. Whether VO_2 NM with lower growth temperature (350 °C and 400 °C) could be completely flat, when the ambient temperature is higher than 80 °C.

Our response:

Thank you for your suggestion. We heat samples i (350 °C) and ii (400 °C) from room temperature to 120 °C, as shown in Re-Fig. 9a, b. It can be clearly seen that samples i and ii complete their morphological changes at τ_c . Further heating treatment can hardly lead to observable morphology change (temperature ≥ 50 °C in Re-Fig. 9a, b). This is consistent with previous report in the literature (DOI: 10.1021/nl303405g and 10.1002/adma.201304064). It is worth noting that the samples can hardly unroll to flat state during heating, suggesting that the strain gradients in the NMs do not completely disappear during phase transition, but are partially offset by the new strains introduced due to phase transition.

Re-Figure 9. a Microscope images of sample i during heating. **b** Microscope images of sample ii during heating.

Our modification:

1. Re-Fig. 9 has been added as supplementary Fig. 19.
2. In addition, it is found that continuous heating of tens of degrees did not lead to observable structural change after the phase transition, indicating that the deformation of the rolled-up structure was almost independent of thermal expansion (supplementary Fig. 19).

Comment:

8. On the other hand, whether VO₂ NM with higher growth temperature (450 °C, 500 °C and 550 °C) bends in the opposite direction, when the ambient temperature is higher than 80 °C.

Our response:

Thank you for your suggestion. We heat samples iii (450 °C), iv (500 °C), and v (550 °C) from room temperature to 120 °C and typical results are shown in Re-Fig. 10a, b. It can be clearly seen that samples iii and v unroll to flat status at τ_c , and the structure no longer changes with continuous heating. According to Fig. 3e in the manuscript, the rolled-up structure can reach its maximum deformation after the phase transition when the strain change from the phase transition is greater than the strain required for rolling. Therefore, sample iii, where the strain required for rolling is close to the phase transition strain, exhibits the maximum deformation — from rolling state to a flat state. For the samples iv and v, the phase transition strain is larger than the rolling strain, and the structure should deform as rolling in the opposite direction after phase transition. However, this is unlikely to be observed as the barrier effect of the substrate.

Re-Figure 10. a Microscope images of sample iii during heating. **b** Microscope images of sample v during heating.

Our modification:

1. Re-Fig. 10 has been added as supplementary Fig. 19.
2. In addition, it is found that continuous heating of tens of degrees did not lead to observable structural change after the phase transition, indicating that the deformation of the rolled-up structure was almost independent of thermal expansion (supplementary Fig. 19).

Comment:

9. On page 12, paragraph 1, the authors wrote that “Moreover, highly controllable K and L^* can modulate ΔT_{sol} from 0.12 (bule region) to 0.57 (red region)”. “bule region” may be the wrong spelling.

Our response:

Sorry for the mistake. We have fixed it.

Our modification:

1. Moreover, highly controllable K and L^* can modulate ΔT_{sol} from 0.12 (blue region) to 0.57 (red region).

Comment:

10. The feature comparison of T_{lum} and ΔT_{sol} between different types of VO₂-based smart window in Fig. 4e is meaningful. But, in my opinion, another more comprehensive feature comparison of T_{lum} and ΔT_{sol} between various smart windows composed of different materials (such as hydrogel, liquid crystal, phase-changing polymer and perovskite) is more important. This additional comparison, showing the advances in a broader field, is helpful to estimate the significance of this work. In addition, what is the potential of VO₂ in smart window, taking T_{lum} and ΔT_{sol} into account?

Our response:

We gratefully thank the reviewer for the constructive remarks. We agree that we should consider more other materials applied in SW for finding the advantage of rolled-up SW. We all know that ΔT_{sol} of conventional VO₂ SWs is regulated by huge change in infrared transmission during the phase transition, but it has few adjustability for luminous transmission. Therefore, our initial aim is to use rolled-up technology, which is compatible to different films, to retain the smart properties of the film while improving its visible light transmittance as much as possible. This is confirmed by the comparison of rolled-up SW with the VO₂ based SWs in Fig. 4e of manuscript. Here, a more comprehensive comparison is demonstrated. Re-Fig. 11a shows a comparison of our work with different materials-based SW works regarding the average T_{lum} and ΔT_{sol} . Hydrogels, ionic liquids, and phase change polymers-based SW have achieved very high ΔT_{sol} , but their T_{lum} at high τ is approaching 0 which means they are loss of lighting capacity. Therefore, obtaining high ΔT_{sol} means minimising light transmission at high temperatures. It is further supported by Re-Fig. 11b, where we compare $\Delta T_{sol} / \Delta T_{lum}$ versus ΔT_{sol} for different materials (a larger $\Delta T_{sol} / \Delta T_{lum}$ means increasing ΔT_{sol} and smaller change in visible light transmission of the two statuses). It can be seen that the normal SW which modulates the ΔT_{sol} by changing visible transmission will have relatively low $\Delta T_{sol} / \Delta T_{lum}$. This comparison highlights the advantages of VO₂ SWs that improves the ΔT_{sol} without changing the lighting. Therefore, many of high ΔT_{sol} SWs are not single material systems, and they combine high visible light transmission materials with VO₂,

expecting better performance through dual modulation in the visible and infrared, such as hydrogel/VO₂ (DOI: 10.1039/c4ta05035e) and ionic liquid/VO₂ (DOI: 10.1021/acsami.6b11202). Similarly, our rolled-up technology can be combined with different material systems to achieve further improvements on the performance of SW. In Re-Fig. 11c and d, the rolled-up technique is supposed to be applied in different material systems (the calculation method is shown in revised supplementary Note 5). One can see that the performance of SW can be further enhanced compared with corresponding planar references. Performance improvement rates of rolled-up SWs made from different materials are shown in Re-Fig. 11e and f. The improvement rates for both ΔT_{sol} and T_{lum} in this work are much higher than those for other materials. The results prove the advantages of the rolled-up technique in improving the performance of planar SWs, especially for films that are inherently with poor visible light modulation, or for planar structures that need to improve visible light transmission at low temperature. In addition to the advantages of VO₂ being able to combine with other materials to achieve a greater range of solar regulation, the light control characteristics of the VO₂ phase transition can compensate the difficulty of adjusting the functionality of radiative cooling SWs to the climate (DOI: 10.1126/science.abg0291). We believe that fabrication of a deformable radiative cooling SW with VO₂ incorporated can further enhance the energy management of the house — The radiative cooling layer can be taken away to the surface of the window by rolling to stop cooling at low τ and automatically re-work at high τ due to strain-induced geometrical recovery by VO₂ phase transition).

Re-Figure 11. **a** Comparison of this work with the best-reported experimental works regarding the average T_{lum} and ΔT_{sol} . **b** Comparison of this work with the best-reported experimental works regarding the ΔT_{sol} and $\Delta T_{sol} / \Delta T_{lum}$. **c** Comparison of this work with simulated ΔT_{sol} enhancement due to rolling. **d** Comparison of this work with simulated T_{lum} due to rolling. **e** Comparison of ΔT_{sol} improved rates of this work and simulated other rolled-up structures made from different materials. **f**

Comparison of T_{lum} improved rates of this work and simulated other rolled-up structures made from different materials.

Our modification:

1. Re-Fig. 11 has been added as supplementary Fig. 29.
2. More, the comparison of T_{lum} and ΔT_{sol} of this work to different materials-based SW is shown in supplementary Fig. 29a, b. Hydrogels, ionic liquids, and phase change polymers-based SW have achieved very high ΔT_{sol} , but their T_{lum} at high τ is approaching 0, suggesting they are loss of light capacity. Interestingly, many of high ΔT_{sol} SWs are not single material systems, they combine high visible light transmission materials with VO₂, expecting better performance through dual modulation in the visible and infrared regions, such as hydrogel/VO₂ and ionic liquid/VO₂. Our rolled-up technology with material compatibility can be combined with different material film systems to achieve further improvements on the performance of SWs (supplementary Fig. 29c-f). Therefore, the current approach can further break through the performance limitations imposed by flat structures.

Reviewer #2

The manuscript is a valuable contribution to the area of micro-mechanic actuators applied to the energy field, as it presents a quite curious new system composed of arrays of self-actuating micro-curtains with outstanding thermochromic performance for smart windows, but with interest in other applications (protective covers for lenses, etc) as proposed by the authors.

The high level of this study and results presented herein, as well as the overall text quality, justify the acceptance of this work in a high-impact journal such as the present one, given that the issues indicated below are properly addressed.

Our response:

Thank you for your comments and constructive advices, which gave us a lot of inspirations. We followed your suggestion and made a careful revision.

Comment:

1. My main concern lies in the scalability of this technology to address the intended application. For practical building integration it would be necessary to pattern several m² of these structures in a timely and inexpensive fashion, which is not at all possible with the photolithography approach employed here. However, there is presently a wide range of soft-lithography methods that should be considered in view of allowing large-area long-range micro-patterning, such as those reported in related works (e.g. doi.org/10.3390/nano11071665 and doi.org/10.1002/admi.202000264). In view of this, the future adaptability of this solution to allow industrial-level scalability is a crucial aspect that needs to be discussed in the manuscript.

Our response:

Thank you for your advice. As we know, the inability to pattern large area with micron scale accuracy is a major constraint to the widespread use of chips. Rolled-up SWs have the same problem, even if they only require a single lithography process. Colloidal

lithography (CL) is considered the preferential structuring method for pattern, as it is an inexpensive and highly scalable soft-patterning technique allowing nanoscopic precision over indefinitely large areas. Tuning specific parameters, such as the size of colloids, shape, monodispersity, and final arrangement, CL enables the production of various structures for different purposes and applications. This provides us with valuable ideas for the future industrial preparation of large-area rolled-up SWs. We have added corresponding discussion in the revised manuscript.

Our modification:

1. In addition, for practical applications, industrial-level scalability is crucial. In our present work, the size of sample used for demo is $\sim 1 \text{ cm}^2$, but we should stress that the fabrication process with photolithography step can prepare patterned VO₂ NMs of larger area (supplementary Fig. 32). To produce rolled-up SWs with even larger size (e.g., $\sim \text{m}^2$ scale in practice), industrial grade magnetron sputtering and inexpensive soft lithography technique, which allows nanoscopic precision over indefinitely large areas,^{59,60} is promising, but more research inputs are still needed for industrial fabrication.

Comment:

2. Another critical aspect which is insufficiently addressed is concerned with the stability of the micro-structured membranes upon repeated cycles of rolling and unrolling. Namely, how does the thermo/mechano-chromic performance decrease with the number of rolling cycles and, from that, can the authors estimate the operational lifetime of this technology in typical conditions?

Our response:

Thank you for your suggestion. The stability of the structure and the cyclability of the shape deformation is one of the most important factors affecting the lifetime of a SW. Figure 4d in the manuscript shows a cycle test of a rolled-up SW during heating and cooling, and the rolled-up VO₂ NMs statistically maintains stable shape deformation for up to 1500 cycles. Using Shanghai, China as a reference, typically it requires the deformation of SW to regulate the room temperature for approximately 1/4 of the year,

and thus the lifetime of a rolled-up SW is about 9 years. To further determine the effect of prolonged storage on the thermo/mechano-chromic performance of rolled-up SW, we performed several heating and cooling cycles again 9 months after the previous cycle test, and the results show excellent deformation stability (Re-Fig. 12a). After 1600 cycles, the rolled-up SW was re-run for light transmission. As shown in Re-Fig. 12b, the results are similar to the earlier ones, and the deviation is about 0.5% in visible light and about 3% in infrared. These results show that the rolled-up SW has good structural stability and deformation recyclability, and it can be used efficiently for long periods of time.

Re-Figure 12. a Fatigue tests of the rolled-up SWs switching between low τ and high τ measured with a time interval of 9 months. Insets are the microscope images of rolled-up SWs at low τ and high τ . Scale bar: 100 μm . **b** Transmittance spectra of the sample V after fatigue tests.

Our modification:

1. Re-Fig. 12a replaces the Fig. 4d in the manuscript.
2. Re-Fig. 12b has been added as supplementary Fig. 25.
3. Furthermore, the rolled-up SW is stable during switch between the two statuses, and few failures occur in 1600 heating-cooling circles, 9 months interval (Fig. 4d). The rolled-up SW after cycle test and prolonged storage also has efficient light modulation ability (supplementary Fig. 25).

Comment:

3. *An energy balance study should also be presented, indicating the power savings that can be expected in average buildings from integrating Smart Windows with this technology, as compared to other SW solutions under development.*

Our response:

Your suggestion is helpful to us. In order to demonstrate more effectively the advantages of rolled-up SWs in practical applications, we set the film to achieve a τ_c of 24°C through doping when studying the energy balance. A house model was built and the house parameters are listed in Re-table 1. The optical data of rolled-up SW, VO₂ NM SW, and normal glass (Re-table 2) were imported into EnergyPlus to simulate the annual energy consumption of the house with the three types of windows in 7 cities, as shown in Re-Fig. 13a. It can be seen that in warmer regions, such as Shanghai, Hong Kong, Albuquerque, and Seattle, the total annual energy consumption of SWs is lower than that of normal glass. But in colder regions, such as Beijing, Moscow, and Whitehorse, the total annual energy consumption of VO₂ NM SW is high than that of normal glass and rolled-up SW due to its poor visible transmission at low τ . For better comparison of rolled-up SW and VO₂ NM SW, energy conversation relative to ordinary glass is shown in Re-Fig. 13b. In this figure, the blue region indicates that annual average temperature is lower than τ_c , while red region is opposite. The calculated results clearly show that the rolled-up SWs provide energy savings in 7 cities whatever the climates are. The energy saving rate of rolled-up SW is compared with other existing SWs reported in Re-table 3. The comparison in the three cities (Hong Kong, Shanghai, and Albuquerque, the three most common cities in the simulation) shows the energy saving rate of rolled-up SW is basically at a high level compared to other SWs based on different materials with high ΔT_{sol} .

Re-Figure 13. **a** Detailed total energy consumption (include cooling, heating, and lighting) in the building. **b** Annual energy conservation with different windows compared to the normal glass window in 7 cities.

Re-Table 1 Building parameters used in the EnergyPlus simulation.

Building type	Small office building
Number of Floors	1
Toral Floor Area	20m × 10m
Average Window-to-Wall Ratio	30%
Temperature setting point for HVAC control	Below 21°C for heating/Above 24°C for cooling
Setpoint for lighting on	1000 lux

Re-Table 2 Optical information of the window used in the simulation.

	Normal window	VO ₂ NM		Rolled-up SW	
		Cold state	Hot state	Cold state	Hot state
States ($\tau_c=24^\circ\text{C}$)	-	Cold state	Hot state	Cold state	Hot state
Solar transmittance	0.775	0.3527	0.2347	0.724	0.2347

Front side solar reflectance	0.071	0.3112	0.2691	0.219	0.2691
Back side solar reflectance	0.071	0.2357	0.1742	0.239	0.1742
Visible transmittance	0.881	0.3247	0.3038	0.722	0.3038
Front side visible reflectance	0.08	0.194	0.1735	0.181	0.1735
Backside visible reflectance	0.08	0.0829	0.0893	0.16	0.0893
Infrared transmittance	0	0	0	0	0
Front side infrared emissivity	0.84	0.823	0.706	0.792	0.706
Back side infrared emissivity	0.84	0.84	0.84	0.84	0.84

Re-Table 3 Annual energy saving of different thermochromic SW in Hong Kong, Shanghai, and Albuquerque, respectively.

Thermochromic SW (window-to-wall ratio)	Hong Kong		Shanghai		Albuquerque	
	$E_c(\text{MJ}/\text{m}^2)$	$E_{\text{saving}}(\%)$	$E_c(\text{MJ}/\text{m}^2)$	$E_{\text{saving}}(\%)$	$E_c(\text{MJ}/\text{m}^2)$	$E_{\text{saving}}(\%)$
Rolled-up SW (0.3)	48.05	10.56	26.10	5.69	71.05	13.08
Perovskite/Low E glass (0.375)	27.65	9.1				
Perovskite/Hydrogel (0.385)	25.65	9.2				
Phase change polymer (0.5)	34.09	13.8				

Hydrogel (0.5)		150					
Liquid hydrogel (0.5)	1 mm			14.73	10.7		
	1 cm			25.68	19.2		
VO₂ radiation cooling (0.33)							6.5

Reference:

- [1] Liu, S. et al. Near-infrared-activated thermochromic perovskite smart windows. *Adv. Sci.* **9**, 2106090 (2022).
- [2] Meng, Y. et al Building-integrated photovoltaic smart window with energy generation and conservation. *Appl. Energy* **324**, 119676 (2022).
- [3] Li, D. et al. Deformable thermo-responsive smart windows based on a shape memory polymer for adaptive solar modulations. *ACS Appl. Mater. Interfaces* **13**, 61196-61204 (2021).
- [4] Lin, C. et al. All-weather thermochromic windows for synchronous solar and thermal radiation regulation. *Sci. Adv.* **8**, eabn7359 (2022).
- [5] Zhou, Y. et al. Liquid thermo-responsive smart window derived from hydrogel. *Joule* **4**, 2458-2474 (2020).
- [6] Ke, Y. et al. On-demand solar and thermal radiation management based on switchable interwoven surfaces. *ACS Energy Lett.* **7**, 1758-1763 (2022).

Our modification:

1. Re-Fig. 13a has been added as supplementary Fig. 31.
2. Re-Fig. 13b has been added as Fig. 5f in the manuscript.
3. Re-table 1-3 have been added as supplementary table 5-7.
4. For further research the energy balance of house installed with SW, a building model was built and EnergyPlus simulation was carried out (the building model and simulation parameters are listed in supplementary table 5 and 6). The obtained annual energy consumptions of houses installed with SWs and normal glass in different cities are shown in supplementary Fig. 31, and the calculated saved energies are shown in Fig. 5f, where the blue areas indicate that the annual average temperature of the city is lower than the τ_c , and the red areas are the opposite. The simulation results visually show that the energy conservation of VO₂-based SWs is more efficient in locations with high average annual temperatures. Comparing the energy conservation of rolled-up SW with VO₂ NM SW, we notice that the rolled-up SW can save energy in any cities with different climates, while VO₂ NM SW is

not suitable for cities with more low-temperature days. Moreover, the energy saving rate of rolled-up SW is also at a high level compared to SWs based on different materials with high ΔT_{sol} (supplementary table 7), indicating the universality of rolled-up SWs for future house windows.

Comment:

4. Lastly, it is not clear why the present structures are named “origami”, since origami usually involve more complex shapes. I recommend re-naming the NMs to a term that is more specific to its actuation, such as “rolling scrolls” or “self-rolling micro-curtains” or something of that sort.

Our response:

Thank you for your suggestion. We have uniformly changed to self-rolling.

Our modification:

1. The article title is change to “Self-rolling of vanadium dioxide nanomembranes for enhanced multi-level solar modulation”.
2. Here, we demonstrated self-rolling of vanadium dioxide (VO₂) nanomembranes (NMs), and designed a smart window (SW) combining mechanochromism with thermochromism to enhance multi-level solar modulation.
3. This work supports the feasibility and superiority of rolled-up technology in SWs and lens protection, which promises broad interest and practical applications of self-adapting devices and systems for smart building, intelligent sensors, and actuators with the perspective of energy efficiency.
4. Here, integrated rolled-up VO₂ SWs integrated VO₂ SWs with high T_{lum} and ΔT_{sol} are realized by the combination of mechanochromism and thermochromism.
5. This research proves that self-rolling can bring improvement to SWs and lens protection, and applies in various passive self-adapting devices and systems.
6. The τ_c of the rolled-up SW reduces consider to be due to the released strain of VO₂ NM in rolling geometry³⁷.

Reviewer #3

In this paper, authors present interesting and unique combination of mechanochromism and thermochromism in the multifaceted vanadium dioxide. The manuscript has applied focus in smart windows and camera shield for self-regulated temperature control. This work provides detail information on the material, thermal, optical, and mechanical characterisations, and fabrication approach. The provided information is enough to reproduce the results. The work is of significant importance to the current hot topics in vanadium dioxide for smart windows and origami-kirigami methods. While the evidences and conclusions are presented well, some more information on the following aspects would benefit the work. I would suggest a minor revision for this work.

Our response:

We appreciate your kind suggestions and we have amended the manuscript accordingly. In addition, the followings are answers for your questions.

Comment:

1. What is the size of the sample (not single flap) authors fabricated to show the proof-of-concept of this? As the focus of this work is applied, how do authors plan do translate this work for large windows? As sputtering has been used here, this deposition method is not very appropriate for large scale.

Our response:

We appreciate for your valuable comment. In our experiment, a 1 cm² sample is covered with an array of rolled-up structures. This size was chosen mainly due to limitation of the HF etching step in fabrication process (we are unable to carry out extensive HF corrosion work in lab.) and drying step in a supercritical dryer (2-inch chamber). In fact, the maximum area of VO₂ prepared by magnetron sputtering in our lab is 4 inches, and the lithography process here also can pattern up to 4-inch samples. Re-Fig. 14a shows a patterned VO₂ NM and an unpatterned VO₂ NM with size of 4 inches, and the

patterned NM has a very significant increase in light transmission. This is also illustrated in Re-Fig. 14b and c. Of course, this is still some way off for application in practical SWs, but industrial grade magnetron sputtering can prepare samples with the size of a standard window, and new photolithography and colloidal lithography (DOI: 10.3390/nano11071665 and 10.1002/admi.202000264) approaches can already fabricate larger patterned samples. We will also be working on industrializing this technology in the coming work, using the industrial grade patterning environment.

Re-Figure 14. **a** Comparison of patterned VO₂ NM wafer (4-inch) with unpatterned VO₂ NM wafer (4-inch) in transparency. **b** Photograph of patterned VO₂ NM wafer (4-inch). **c** Photograph of unpatterned VO₂ NM wafer (4-inch).

Our modification:

1. Re-Fig. 14 has been added as supplementary Fig. 32.
2. In addition, for practical applications, industrial-level scalability is crucial. In our present work, the size of sample used for demo is $\sim 1 \text{ cm}^2$, but we should stress that the fabrication process with photolithography step can prepare patterned VO₂ NMs of larger area (supplementary Fig. 32). To produce rolled-up SWs with even larger size (e.g., $\sim \text{m}^2$ scale in practice), industrial grade magnetron sputtering and inexpensive soft lithography technique, which allows nanoscopic precision over

indefinitely large areas,^{59,60} is promising, but more research inputs are still needed for industrial fabrication.

Comment:

2. The typical transition temperature for VO₂ is 68C. More information and relevant references should be provided in the paper to reason why authors are getting transition at lower temperature of 48C

Our response:

Thank you for your suggestion. Here, the phase transition temperature (τ_c) decreases because VO₂ NM receives a compressive strain after being released from the substrate (DOI: 10.1021/acs.nanolett.8b00483). Reason for this strain generation is explained in Re-Fig. 7. Due to oxygen insufficiency during the initial growth stage, a buffer layer was formed corresponding to the formation of oxygen-deficient Magnéli phases V_nO_{2n-1} (DOI: 10.1063/1.3642980). This buffer layer is still dominated by elemental V and O, and hardly affects the VO₂ NM components and crystalline quality. The lattice mismatch between the upper pure VO₂ NM and buffer layer creates a stress gradient in the vertical direction. The initial stress gradient in the VO₂ NM can be demonstrated by the XRD results of the synchrotron radiation variation angle (supplementary Fig. 5 in revised Supplementary Information) and the reduction of τ_c is also demonstrated in the variable temperature Raman results in Re-Fig. 8. To further demonstrate the strain generation by rolling, we compared the Raman spectra of V1, V2 and O peaks of samples iii, iv, v and VO₂ NM under 633 nm laser (Re-Fig. 15a) and the peak shifts are summarized in Re-Fig. 15b. It can be seen that the changes of strain lead to a monotonic peak shift, and this result agrees well with previous literature (DOI: 10.1103/PhysRevB.85.020101 and 10.1021/acs.nanolett.8b00483).

Re-Figure 15. **a** Raman spectra of sample iii, iv, v and VO₂ NM excited by lasers with wavelengths of 633 nm at room temperature. **b** Peak positions (ω_{v1} , ω_{v2} , and ω_o) determined from Lorentzian fitting of the spectra shown in (a).

Our modification:

1. Re-Fig. 15 has been added as supplementary Fig. 22.
2. The τ_c of the rolled-up SW reduction considers to be due to the released strain of VO₂ NM in **rolling geometry**³⁷.

Comment:

3. More information should be provided for Figure. 3b statistical data. Especially how many samples are tested to calculate the error graph for each temperature.

Our response:

We apologize for not providing detailed information. Re-Fig. 16-20 are the overall or partial photos of rolled-up SWs and curvature (K) statistics of rolled-up structures with various K at different temperatures. About 1600 rolled-up units were counted for sample i and about 700 rolled-up units were counted for samples ii, iii, iv, and v. The errors in Fig. 3b of manuscript in the manuscript were calculated from these results.

Re-Figure 16. **a** Photograph (left) and corresponding optical microscope image (right) of sample i at low τ . **b** Statistics of sample i at low τ . **c** Photograph (left) and corresponding optical microscope image (right) of sample i at high τ . **d** Statistics of sample i at high τ .

Re-Figure 17. **a** Photograph (left) and corresponding optical microscope image (right) of sample ii at low τ . **b** Statistics of sample ii at low τ . **c** Photograph (left) and

corresponding optical microscope image (right) of sample ii at high τ . **d** Statistics of sample ii at high τ .

Re-Figure 18. a Photograph (left) and corresponding optical microscope image (right) of sample iii at low τ . **b** Statistics of sample iii at low τ . **c** Photograph (left) and corresponding optical microscope image (right) of sample iii at high τ . **d** Statistics of sample iii at high τ .

Re-Figure 19. **a** Photograph (left) and corresponding optical microscope image (right) of sample iv at low τ . **b** Statistics of sample iv at low τ . **c** Photograph (left) and corresponding optical microscope image (right) of sample iv at high τ . **d** Statistics of sample iv at high τ .

Re-Figure 20. **a** Photograph (left) and corresponding optical microscope image (right) of sample v at low τ . **b** Statistics of sample v at low τ . **c** Photograph (left) and corresponding optical microscope image (right) of sample v at high τ . **d** Statistics of sample v at high τ .

Our modification:

1. Re-Fig. 16-20 has been added as supplementary Fig. 10-14.
2. Control the NM at the same thickness by the same deposition time, the rolled-up SW samples deposited at different temperatures (i, ii, iii, iv, and v) demonstrate various initial K and deformations ΔK in MIT (Fig. 3b, detailed experimental results in supplementary Fig. 10-14 and calculation method of error bar is SD).

Comment:

4. *Is this un-rolling effect is gradual or occurs suddenly at transition temperature? What is the rolling recover time? The images provided in the paper are only at 20C and*

80C. Can authors provide images at intermittent temperatures as well? It would be interesting to see if the rolling is caused by thermal stress or it is triggered by the phase-change.

Our response:

The phase transition of VO₂ is an ultrafast process, taking only ms or even ps to complete when the temperature is above τ_c (DOI: 10.1126/science.1253779 and 10.1038/s41467-020-20843-4). The actuation of the rolled-up structure in our work relies on the phase transition, so the actuation process is also a sudden change in a short time period, and the length of time depends mainly on how fast the temperature increases or decreases, which is confirmed by our experiments and references (DOI: 10.1038/s41467-020-20843-4 and 10.1021/acs.nanolett.8b00483). In our work, we used a high-speed camera to record the actuation process of a laser-heated rolled-up structure, as shown in Re-Fig. 21, which took 6.7 ms from the laser “on” to the completion of the rolling process, and 4 ms to for the structure to recover its original shape after the laser was turned off. In addition, a rolled-up SW was placed on the heating plate for overall heating and the photos of the heating process and recovery process are recorded in Re-Fig. 22. It shows that different temperatures correspond to different K within temperature range of 40 -50 °C. We further investigated the change of K during heating and cooling (Re-Fig. 23). Re-Fig. 23a shows the K versus time curve for a heating rate of 0.1 °C/s, and the deformation time is about 102 s. When the heating rate is increased to 0.4 °C/s, the deformation time decreases to 21 s (Re-Fig. 23b). Re-Fig. 23c shows the stepwise heating process, start from 40 °C and wait 10 s for every 2 °C step. It is found that when the temperature stabilizes at any temperature between 40 °C and 50 °C, the rolled-up SW demonstrates a stable K . Reason for this phenomenon is that the phase transition of VO₂ is a process in which R-phase region gradually increases and M-phase region gradually disappears, and there is a state of co-existence of the two phases in the temperature range close to τ_c (DOI: 10.1021/nl500042x and 10.1021/nl900841k). During the natural cooling process,

deformation of rolled-up structure started at 40 °C and the structure restore the original K at 34 °C. The recovery process taking approximately 122s (Re-Fig. 23d). These results show that although the deformation of the rolled-up structure is abrupt after the ambient temperature reaches the triggering temperature, the actual temperature change is slower leading to a gradual deformation.

Re-Figure 21. Side views of 808 nm laser-excited rolled-up structure with phase transition.

Re-Figure 22. Microscope images of sample iii at different temperatures.

Re-Figure 23. **a** K of sample iii as a function of time in heating process ($0.1 \text{ }^{\circ}\text{C/s}$). **b** K of sample iii as a function of time in heating process ($0.4 \text{ }^{\circ}\text{C/s}$). **c** K of sample iii as a function of time in stepwise heating process. **d** K of sample iii as a function of time in cooling process.

Our modification:

1. Re-Fig. 21 has been added as supplementary Fig. 17.
2. Re-Fig. 22 has been replaced Fig. 3c in the manuscript.
3. Re-Fig. 23 has been added as supplementary Fig. 18.
4. On the other hand, when the temperature is reduced to $34 \text{ }^{\circ}\text{C}$, the rolled-up structure returned to its initial geometry. The response time of the actuation is ultrafast at τ_c : a rolled-up structure can complete the deformation in 6.7 ms stimulated by laser heating (supplementary Fig. 17). For overall heating, the time required for deformation is closely related to the variation in temperature (supplementary Fig. 18).

Comment:

5. Is there any relationship between the effect of K and L/L^* modulation on critical transition temperature?

Our response:

Thank you for your interesting question. As we mentioned in the previous comment 2, the main reason for the lower phase transition temperature (τ_c) in rolled-up structures is the internal strain. A different K means that the rolled-up structures have different strain gradients, with a larger K resulting in a higher strain, more lattice compression and a lower τ_c . As shown in the K versus temperature curve (Re-Fig. 24a), the point of maximum slope of the curve (i.e., τ_c) differs significantly in different samples. Statistics in Re-Fig. 24b shows that the rolled-up structure with the smaller K has a higher τ_c (sample i has a deviation in τ_c due to the poor crystal quality and impure composition caused by growth of VO_2 NM at low temperature). On the other hand, change of L/L^* suggests the tuning of the spacing between the rolled-up structures, as shown in Re-Fig. 25, and the rolled-up structures have same K , indicating the same strain in these samples. Therefore, τ_c of these samples are same.

Re-Figure 24. **a** K of different rolled-up VO_2 samples as a function of temperature. **b** Statistics on τ_c , derived from (a).

Re-Figure 25. Microscope images of sample II, IV and V.

Comment:

6. *Typo – Line 331 and 332 page 16 – Should it be 12 pm? And not 12 am.*

Our response:

Thank you for your comment. We have carefully checked the manuscript and fixed the typos.

Our modification:

1. The temperature of rolled-up SW-installed house before 12:00 pm was between the other two houses, which is similar to the situation in winter. After 12:00 pm, significant thermal insulation effect of the rolled-up SW beneficial from shape deformation because the phase transition occurred.

Reviewer #4

In the manuscript, the authors reported the vanadium dioxide (VO₂)-based three-dimensional (3D) smart windows (SWs) combining mechanochromism with thermochromism to enhance multi-level solar modulation. Due to the rolling origami of VO₂ NMs integrating on quartz, the 3D SW was able to synergy of shape deformation and light transmission when the phase transition temperature was reached, which significantly increased the solar modulation and luminous transmittance. The manuscript also presented impressive experimental data and careful data analyses to extract the principle of control the deformation with temperature in a VO₂-based microstructure. This strategy shown in the paper provided a new idea for further energy efficiency improvements in SWs based on planar structures. Therefore, this manuscript should be recommended for publication in Nature Communications.

Our response:

We highly appreciate your positive evaluation on our work. We responded the concerns pointed out by the reviewer and made a careful revision.

Comment:

1. It was known that a crystal structural variation was vital in the VO₂ phase transition, but it was difficult to see the huge difference in the position and arrangement of the atoms in the TEM images. So how should the completion of the phase transition be judged in the in-situ TEM measurement?

Our response:

Thank you for your comments. The atomic-resolution STEM image of VO₂ NM and corresponding fast Fourier transformation (FFT) are concordant with the [010] zone axis of the monoclinic structure at different temperatures (Re-Fig. 26a-f). The bright dots with similar intensity in STEM images represent projected V atoms. According to

the intensity of V atomic column, the interplanar spacing (d) in different zone axis is shown in the STEM image and an angle of 87.6° between different crystalline planes is noticed. In order to resolve the accurate atomic arrangement and crystal information, we built a crystal model based on the crystal file (mp-102155) and found that the model fits well with the inverse FFT image. It indicates that the bright dots in the STEM image is the overlap of two V atoms, but the values of d can still be trusted due to the uniform distribution between the V atoms. As the temperature increases to 80°C , the number of bright spots in the FFT decreases, while the d in different zone axis has significant changes and the angle between two crystalline planes changes to 78.2° . These crystal parameters in 80°C correspond to the metallic O phase of VO_2 (mp-1094031) which proves the completion of the phase transition. Moreover, the regular variation of the d with temperature represented in Re-Fig. 26g also shows the process of phase transition.

Re-Figure 26. In-situ STEM image and corresponding FFT image of VO_2 NM during heating. **a** 20°C , **b** 50°C , **c** 80°C , **d** 110°C , **e** 140°C , and **f** 170°C . **g** Interplanar spacing (d) in two directions changes with temperature.

Comment:

2. In Fig. 3b, the authors mentioned that the relatively small deformation of the low temperature grown VO_2 NMs during the phase transition was due to the poor NMs quality and the impure composition. For this concern, my suggestion was to add some supporting tests to prove this point, such as XRD, Raman, etc.

Our response:

Thank you for your suggestion. Re-Fig. 27 shows that the VO₂ NMs with low growth temperature have relatively weak Raman peak. In contrast, the relative peak intensities improve significantly when the growth temperature is increased to 450 °C. This proves the poor crystalline quality of the VO₂ NM deposited at low temperature.

Re-Figure 27. Raman spectra of VO₂ NM samples deposited at different temperatures excited by laser with wavelength of 532 nm.

Our modification:

1. Re-Fig. 27 has been added as supplementary Fig. 16.
2. ΔK of sample i and ii are smaller than those of the other samples due to their low crystallinity and impure ingredients caused by depositing at low growth temperatures (supplementary Fig. 16).

Comment:

3. *The authors focused on improving the solar modulation by varying the radius and the structures alignment period (L/L^*). The L/L^* could theoretically be further increased to 1 for a better solar modulation. Did the authors attempt to further increase L/L^**

Our response:

We agree with the comment about the further optimization. We have also tried to increase L/L^* further, which implies a spacing between rolled-up structures less than 30 μm . As shown in Re-Fig. 28, when the spacing is reduced to 20 μm , the integrity of

the rolled-up SW is difficult to maintain during wet etching process, because the overly narrow framework will also be released before the rolling of VO₂ NM. Therefore, $L/L^* = 0.87$ was chosen for high success rate in current fabrication process. For enhanced solar modulation, we will also look for methods that allow for stable fabrication of larger L/L^* structures in subsequent work.

Re-Figure 28. Microscope image of rolled-up SW with L/L^* of 0.91.

Comment:

4. I suggested that the authors could describe the simulation process in more detail and provide n , k values for the VO₂ NM in low and high temperatures to improve the credibility of the simulation results.

Our response:

Thank you for your suggestion. The transmittance simulation was performed by using commercial software Essential Macleod (Thin Film Center Inc, USA) based on the multilayer optical theory with a transfer-matrix method. The optical constants (n , k) of VO₂ NM in low and high temperatures were taken from literature (DOI: 10.1021/acsami.9b03586) in the range of 350-2500 nm (Re-Fig. 29). Refractive index of the quartz substrate was set to be 1.5. The thickness of VO₂ NM was fixed at 100 nm when simulating the spectra transmittance $T(\lambda)$ at low and high temperatures.

At high temperature, rolled-up SW maintained flat on the quartz substrate, and its transmittance $\bar{T}_{sol,high} \tau_{rolled-up}$ is approximately equal to the transmittance of the VO₂ NM $\bar{T}_{sol,high} \tau_{NM}$:

$$\bar{T}_{\text{sol,high } \tau_{\text{rolled-up}}} = \bar{T}_{\text{sol,high } \tau_{\text{NM}}}$$

At low temperature, the rolled-up SW shifts to rolled state, and the relationship between the transmittances and the geometric parameters can be approximately described as:

$$\bar{T}_{\text{sol,low } \tau_{\text{rolled-up}}} = \bar{T}_{\text{sol,low } \tau_{\text{NM}}} \cdot \left[1 - L \cdot (L - 1/k)/L^*{}^2 \right] + L \cdot (L - 1/k)/L^*{}^2$$

Therefore, solar modulation ability of rolled-up SW can be calculated:

$$\Delta \bar{T}_{\text{sol,rolled-up}} = \bar{T}_{\text{sol,low } \tau_{\text{rolled-up}}} - \bar{T}_{\text{sol,high } \tau_{\text{rolled-up}}}$$

Re-Figure 29. Wavelength dependent refractive index (n) and extinction coefficient (k) of VO₂ NM at different temperatures.

Our modification:

1. A simple model is built to simulate ΔT_{sol} as a function of K and L/L^* (Fig. 4a), and the simulation method is described in supplementary Note 5.

----- End of Responses to comments -----

----- List of Changes (manuscript)-----

1. The article title has been changed to “Self-rolling of vanadium dioxide nanomembranes for enhanced multi-level solar modulation”.
2. In Page 1, Lines 29, the words “rolling origami” have been revised to “self-rolling”.
3. In Page 1, Lines 37, the words “rolling origami” have been revised to “self-rolling”.
4. In Page 1, Lines 42, the words “rolling origami” have been revised to “self-rolling”.
5. In Page 2, Lines 74-75, the words “new” and “novel” have been deleted.
6. In Page 2, Lines 76-77, the sentence “Here, rolling origami integrated VO₂ SWs with high T_{lum} and ΔT_{sol} are realized by the combination of mechanochromism and thermochromism” has been revised to “Here, integrated rolled-up VO₂ SWs with high T_{lum} and ΔT_{sol} are realized by the combination of mechanochromism and thermochromism”.
7. In Page 2, Lines 86, the words “rolling origami” have been revised to “self-rolling”.
8. In Page 6, Lines 137-138, the sentence “In the transmission electron microscopy (TEM) image, the VO₂ NM between Quartz (lower part) and Pt (upper part) possesses orientation in the vertical direction and a buffer layer is noticed near the substrate (Fig. 2b).” has been revised to “In the transmission electron microscopy (TEM) image, the VO₂ NM between Quartz (lower part) and Pt (upper part. Pt layer mainly serves to protect the surface of the VO₂ NM during preparation of the cross-sectional sample.) possesses orientation in the vertical direction and a buffer layer is noticed near the substrate (Fig. 2b)”.
9. In Page 7, Lines 141, the sentence “The difference in components leads to a lattice mismatch between the upper pure VO₂ NM and buffer layer, which creates a stress gradient along the vertical direction for self-rolling (supplementary Fig. 3)” has been added.
10. Figure 3c of manuscript has been revised as follow:

Revise to

11. In Page 10, Lines 200-203, the sentence “Finally, HF solution was used to etch the quartz, which releases the VO₂ NMs for rolling process” has been revised to “Finally, HF solution was used to etch the quartz, which released the VO₂ NMs for rolling process. In order to fix the framework (i.e., un-rolled) regions, photoresist layer might be used to cover these regions which helps to maintain the intact framework geometry (supplementary Fig. 8)”.

12. In Page 10, Lines 208-211, the sentence has been modified: “Control the NM at the same thickness by the same deposition time, the rolled-up SW samples deposited at different temperatures (i, ii, iii, iv, and v) demonstrate various initial K and deformations ΔK in MIT (Fig. 3b)” has been revised to “Control the NM at the same thickness by the same deposition time, the rolled-up SW samples deposited at different temperatures (i, ii, iii, iv, and v) demonstrate various initial K and deformations ΔK in MIT (Fig. 3b, detailed experimental results in supplementary Fig. 10-14 and calculation method of error bar is SD)”.

13. In Page 11, Lines 227-235, the sentence has been modified: “The phase transition is completed when the temperature is increased to 48 °C, and the rolled-up structure is fully expanded and covers the quartz substrate” has been revised to “The phase transition completed when the temperature increased to 48 °C, and the rolled-up structure fully expanded and covered the quartz substrate. On the other hand, when the

temperature is reduced to 34 °C, the rolled-up structure returned to its initial geometry. The response time of the actuation is ultrafast at τ_c : a rolled-up structure can complete the deformation in 6.7 ms stimulated by laser heating (supplementary Fig. 17). For overall heating, the time required for deformation is closely related to the variation in temperature (supplementary Fig. 18). In addition, it is found that continuous heating of tens of degrees did not lead to observable structural change after the phase transition, indicating that the deformation of the rolled-up structure was almost independent of thermal expansion (supplementary Fig. 19)".

14. In Page 11, Lines 235, the words "rolling origami" have been revised to "rolling geometry".

15. In Page 11, Lines 236-239, the sentence has been modified: "Other experimental result that confirms τ_c of sample iii is in-situ heating Raman spectra measurement, and the A_g vibration peak belonging to the $VO_2(M)$ disappears at 50 °C and reappears at < 30 °C" has been revised to "Other experimental result that confirms τ_c of sample iii is in-situ Raman spectra measurement, and A_g vibration peak belonging to the $VO_2(M)$ disappears at 50 °C and reappears during cooling process at < 40 °C (supplementary Fig. 20)".

16. Figure 4d of manuscript has been revised as follow:

Revise to ↓

17. In Page 13, Lines 275-277, the sentence “A simple model is built to simulate ΔT_{sol} as a function of K and L/L^* (Fig. 4a)” has been revised to “A simple model is built to simulate ΔT_{sol} as a function of K and L/L^* (Fig. 4a), and the simulation method is described in supplementary Note 5”.

18. In Page 13, Lines 278, the word “bule” has been revised to “blue”.

19. In Page 14, Lines 299-300, the sentence “Furthermore, the rolled-up SW is stable during switch between the two statuses, and few failures occur in 1500 heating-cooling circles (Fig. 4d)” has been revised to “Furthermore, the rolled-up SW is stable during

switch between the two statuses, and few failures occur in 1600 heating-cooling circles, 9 months interval (Fig. 4d)”.

20. In Page 14, Lines 300, the sentences “The rolled-up SW after cycle test and prolonged storage also has efficient light modulation ability (supplementary Fig. 25). Moreover, in order to better meet the challenges in actual environments, rolled-up SW can be encapsulated by polydimethylsiloxane between two layers of glass (supplementary Fig. 26), and the encapsulated rolled-up SW thus can withstand wind/rain and chemical attacks, and most importantly, maintain efficient light modulation ability by phase transition (supplementary Fig. 27)” have been added.

21. In Page 14, Lines 305-306, the sentence “The excellent shape deformation stability of the rolled-up SW provides the potential for building windows in practical environments” has been revised to “The excellent shape deformation stability and environmental adaptability of the rolled-up SW provides the potential for building windows in practical environments”.

22. In Page 15, Lines 318, the sentences “More, the comparison of T_{lum} and ΔT_{sol} of this work to different materials-based SW is shown in supplementary Fig. 29a, b. Hydrogels, ionic liquids, and phase change polymers-based SW have achieved very high ΔT_{sol} , but their T_{lum} at high τ is approaching 0, suggesting they are loss of light capacity. Interestingly, many of high ΔT_{sol} SWs are not single material systems, they combine high visible light transmission materials with VO₂, expecting better performance through dual modulation in the visible and infrared regions, such as hydrogel/VO₂ and ionic liquid/VO₂. Our rolled-up technology with material compatibility can be combined with different material film systems to achieve further improvements on the performance of SWs (supplementary Fig. 29c-f). Therefore, the current approach can further break through the performance limitations imposed by flat structures” have been added.

23. Figure 5f of manuscript has been added as follow:

Revise to ↓

24. In Page 17, Lines 354, the word “12:00 am” have been revised to “12:00 pm”.
25. In Page 17, Lines 355, the word “12:00 am” have been revised to “12:00 pm”.
26. In Page 17, Lines 364, the sentences “For further research the energy balance of house installed with SW, a building model was built and EnergyPlus simulation was carried out (the building model and simulation parameters are listed in supplementary table 5 and 6). The obtained annual energy consumptions of houses installed with SWs and normal glass in different cities are shown in supplementary Fig. 31, and the calculated saved energies are shown in Fig. 5f, where the blue areas indicate that the annual average temperature of the city is lower than the τ_c , and the red areas are the opposite. The simulation results visually show that the energy conservation of VO₂-based SWs is more efficient in locations with high average annual temperatures. Comparing the energy conservation of rolled-up SW with VO₂ NM SW, we notice that the rolled-up SW can save energy in any cities with different climates, while VO₂ NM SW is not suitable for cities with more low-temperature days. Moreover, the energy saving rate of rolled-up SW is also at a high level compared to SWs based on different materials with high ΔT_{sol} (supplementary table 7), indicating the universality of rolled-up SWs for future house windows. In addition, for practical applications, industrial-level scalability is crucial. In our present work, the size of sample used for demo is $\sim 1 \text{ cm}^2$, but we should stress that the fabrication process with photolithography step can prepare patterned VO₂ NMs of larger area (supplementary Fig. 32). To produce rolled-up SWs with even larger size (e.g., $\sim \text{m}^2$ scale in practice), industrial grade magnetron sputtering and inexpensive soft lithography technique, which allows nanoscopic precision over indefinitely large areas,^{60,61} is promising, but more research inputs are still needed for industrial fabrication” have been added.
27. In Page 18, Lines 410, the words “rolling origami” have been revised to “rolled-up”.
28. References list has been modified and new refereces 37, 61, and 61 have been added.

----- List of Changes (supplementary information)-----

1. The supplementary Figure 3 has been added as follow:

Supplementary Figure 3: **Stress gradients introduced into VO₂ NM.** **a** TEM image of the cross-section of VO₂ NM. **b** Schematic representations of rolling-up processes of single-component VO₂ nanomembranes. The red arrows qualitatively represent contraction of the layers after the NM released. The difference in these components leads to a lattice mismatch between the upper pure VO₂ NM and buffer layer, which creates a stress gradient in the vertical direction. When the NM with the vertical stress gradient is released from the substrate, the larger contraction of the upper part of the NM compared to the lower part creates a net bending force with the direction away from the substrate. At the same time, for NMs with same thickness, different growth temperatures enable the preparation of VO₂ NM with different initial stress gradients due to the difference in thermal expansion coefficients, which leads to formation of rolled-up VO₂ NM with different curvatures.

2. The supplementary Figure 8 has been added as follow:

Supplementary Figure 8: **Fabrication process for SWs with very small intervals in a rolled-up structure.** **a** The schematic of fabrication process using rolled-up nanotechnology. **b** Microscope images of sample with different etching times. In HF etching step, the quartz in the framework regions will also be gradually etched and the VO₂ NMs in the framework regions are also partially release from the substrate. It is

worth noting that the framework NMs are interconnected and therefore rolling is unlikely realized in these framework regions with short etching time. In order to further fix NMs in these regions, in the fabrication process of our experiment, we perform an additional step of lithography before wet-etching to cover the framework with a thick photoresist layer, which helps to maintain the framework geometry. The experimental results also confirm the effectiveness of the method, especially for short etching time. It can be seen that the framework is intact until the etching time reaches 1800 s, which is much longer than normal etching time of ~ 300 s.

3. The supplementary Figure 10 has been added as follow:

Supplementary Figure 10: K statistics for sample i. **a** Photograph (left) and corresponding optical microscope image (right) of sample i at low τ . **b** Statistics of sample i at low τ . **c** Photograph (left) and corresponding optical microscope image (right) of sample i at high τ . **d** Statistics of sample i at high τ .

4. The supplementary Figure 11 has been added as follow:

Supplementary Figure 11: K statistics for sample ii. **a** Photograph (left) and corresponding optical microscope image (right) of sample ii at low τ . **b** Statistics of sample ii at low τ . **c** Photograph (left) and corresponding optical microscope image (right) of sample ii at high τ . **d** Statistics of sample ii at high τ .

5. The supplementary Figure 12 has been added as follow:

Supplementary Figure 12: K statistics for sample iii. **a** Photograph (left) and corresponding optical microscope image (right) of sample iii at low τ . **b** Statistics of

sample iii at low τ . **c** Photograph (left) and corresponding optical microscope image (right) of sample iii at high τ . **d** Statistics of sample iii at high τ .

6. The supplementary Figure 13 has been added as follow:

Supplementary Figure 13: **K statistics for sample iv.** **a** Photograph (left) and corresponding optical microscope image (right) of sample iv at low τ . **b** Statistics of sample iv at low τ . **c** Photograph (left) and corresponding optical microscope image (right) of sample iv at high τ . **d** Statistics of sample iv at high τ .

7. The supplementary Figure 14 has been added as follow:

Supplementary Figure 14: K statistics for sample v . **a** Photograph (left) and corresponding optical microscope image (right) of sample v at low τ . **b** Statistics of sample v at low τ . **c** Photograph (left) and corresponding optical microscope image (right) of sample v at high τ . **d** Statistics of sample v at high τ .

8. The supplementary Figure 16 has been added as follow:

Supplementary Figure 16: **Raman spectra of VO_2 NM samples deposited at different temperatures excited by laser with wavelength of 532 nm.** The VO_2 NMs with low growth temperature have relatively weak Raman peak. In contrast, the relative peak intensities improve significantly when the growth temperature is increased to 450 $^\circ\text{C}$. This proves the poor crystalline quality of the VO_2 NM deposited at low temperature.

9. The supplementary Figure 17 has been added as follow:

Supplementary Figure 17: **Side views of 808 nm laser-excited rolled-up structure with phase transition.** We used a high-speed camera to record the actuation process of a laser-heated rolled-up structure, which took 6.7 ms from the laser “on” to the completion of the rolling process, and 4 ms to for the structure to recover its original shape after the laser was turned off.

10. The supplementary Figure 28 has been added as follow:

Supplementary Figure 18: **K of sample iii in heating or cooling process.** **a** K of sample iii as a function of time in heating process ($0.1\text{ }^{\circ}\text{C/s}$). **b** K of sample iii as a function of time in heating process ($0.4\text{ }^{\circ}\text{C/s}$). **c** K of sample iii as a function of time in stepwise heating process. **d** K of sample iii as a function of time in cooling process. Supplementary Fig.18a shows the K versus time curve for a heating rate of $0.1\text{ }^{\circ}\text{C/s}$, the deformation time is about 102 s. When the heating rate is increased to $0.4\text{ }^{\circ}\text{C/s}$, the deformation time decreases to 21 s (supplementary Fig.18b). Supplementary Fig.18c shows the stepwise heating process, start at $40\text{ }^{\circ}\text{C}$ and wait 10 s for every $2\text{ }^{\circ}\text{C}$ rise. It

is found that when the temperature stabilizes at any temperature between 40 °C and 50 °C, the rolled-up SW is corresponded to a stable K . During the natural cooling process, deformation of rolled-up structure started at 40 °C and the structure restore the original K at 34 °C, and the recovery process taking approximately 122s (supplementary Fig.18d). These results show that although the deformation of the rolled-up structure is abrupt after the ambient temperature reaches the triggering temperature, the actual temperature change is slower leading to a gradual deformation.

11. The supplementary Figure 19 has been added as follow:

Supplementary Figure 19: **Structural changes in rolled-up samples during heating.** **a** Microscope image of sample i during heating. **b** Microscope image of sample ii during heating. **c** Microscope image of sample iii during heating. **d** Microscope image of sample v during heating. Each sample was subjected to continuous heating to 120 °C after the phase transition and it was found that continuous heating did not lead to observable structural changes.

12. The supplementary Figure 20 has been added as follow:

Supplementary Figure 20: **Raman spectra of a rolled-up structure measured at different temperatures.** **a** Raman spectra of a rolled-up structure measured in heating or cooling process excited by laser with wavelength of 532 nm. **b** Raman spectra of a rolled-up structure measured in heating or cooling process excited by laser with wavelength of 633 nm. **c** Raman peak positions (ω_{V1} and ω_{V2}) as a function of temperature (laser: 532 nm). **d** Raman peak positions (ω_{V1} and ω_{V2}) as a function of temperature (laser: 633 nm). Raman peak of VO₂ disappears at 50 °C and appears again while the temperature reduces to 40 °C. It can be visualised that the position of the peaks fluctuates within the tolerance range before the phase transition, indicating a trivial strain change in this stage. However, it is worth noting that characterization of the strain change during the phase transition by Raman is difficult because the metal phase of VO₂ demonstrates no Raman activity.

13. The supplementary Figure 22 has been added as follow:

Supplementary Figure 22: **Raman peak shifts of different samples and Phase diagrams of VO₂.** **a** Raman spectra of sample iii, iv, v and VO₂ NM excited by lasers with wavelengths of 633 nm at room temperature. **b** Peak positions (ω_{V1} , ω_{V2} , and ω_O) determined from Lorentzian fitting of the spectra shown in (a). **c** Phase diagrams of VO₂. To further demonstrate the strain generation by rolling, we compared the Raman spectra of V1, V2 and O peaks of samples iii, iv, v and VO₂ NM under 633 nm laser (supplementary Fig. 22a) and their peak shifts are summarized in supplementary Fig. 22b. It can be seen that the changes of strain lead to a monotonic shift, and this result agrees with previous literature and the strain calculations are close to the actual test results in these articles. In Phase diagrams of VO₂ (supplementary Fig. 22c), the red line is the strain of the sample iii. It shows that the calculated strain value and the actual ϵ_c are consistent with the phase diagram data.

14. The supplementary Figure 25 has been added as follow:

Supplementary Figure 25: **Transmittance spectra of the sample V after fatigue tests.** The rolled-up SW has good structural stability and deformation recyclability, and it can be used efficiently for long periods of time.

15. The supplementary Figure 26 has been added as follow:

Supplementary Figure 26: **Encapsulating of rolled-up SW.** **a** Schematic of an encapsulated device. **b** Encapsulated rolled-up SW. We place the rolled-up SW sample in the center of normal glass, and the sample is surrounded by polydimethylsiloxane (PDMS) as a bracket to support for the upper glass. The PDMS is then cured at high temperature to form a completely sealed cavity, and thus the sample is insulated from water, wind, air, and other factors that can cause damage to the rolled-up structure.

16. The supplementary Figure 27 has been added as follow:

Supplementary Figure 27: **Encapsulated rolled-up SW test in extreme environments.** Encapsulated rolled-up SW and VO₂ NM placed in (a) NaOH and (b) H₃PO₄ solution. **c** Reference pH values. **d** The process of cleaning with flush of water flow between each test. **e** Light transmission modulation ability after acid and alkaline tests. These results indicate that the encapsulated rolled-up SW should be able to meet the requirements of the complex situations in practical environment.

17. The supplementary Figure 29 has been added as follow:

Supplementary Figure 29: **Comparison of this work with different materials-based SW works regarding the T_{lum} and ΔT_{sol}** **a** Comparison of this work with the best-reported experimental works regarding the average T_{lum} and ΔT_{sol} . **b** Comparison of this work with the best-reported experimental works regarding the ΔT_{sol} and $\Delta T_{sol} / \Delta T_{lum}$. **c** Comparison of this work with simulated ΔT_{sol} enhancement due to rolling. **d** Comparison of this work with simulated T_{lum} due to rolling. **e** Comparison of ΔT_{sol} improved rates of this work and simulated other rolled-up structures made

from different materials. **f** Comparison of T_{lum} improved rates of this work and simulated other other rolled-up structures made from different materials. Here, a more comprehensive comparason is demonstrated (supplementary Fig. 29a). Hydrogels, ionic liquids, and phase change polymers-based SW have achieved very high ΔT_{sol} , but their T_{lum} at high τ is approaching 0 which means they are loss of lighting capacity. Therefore, obtaining high ΔT_{sol} means minimising light transmission at high temperatures. We further compare $\Delta T_{sol} / \Delta T_{lum}$ versus ΔT_{sol} for different materials (a larger $\Delta T_{sol} / \Delta T_{lum}$ means increasing ΔT_{sol} and smaller change in visible light transmission during two status). It can be seen that the normal SW which modulates the ΔT_{sol} by changing visible transmission will have relatively low $\Delta T_{sol} / \Delta T_{lum}$ (supplementary Fig. 29b). This comparison highlights the advantages of VO₂ SWs that improves the ΔT_{sol} without changing the lighting. Moreover, our rolled-up technology can be combined with different material systems to achieve further improvements on the performance of SW. In supplementary Fig. 29c and d, the rolled-up technique is supposed to be applied in different material systems (the calculation method is shown in supplementary Note 5). Performance improvement rates of rolled-up SWs made from different materials are shown in supplementary Fig. 29e and f. The improvement rates for both ΔT_{sol} and T_{lum} in this work are much higher than those for other materials. The results indicate the advantages of the rolled-up technique in improving the performance of planar SWs, especially for films that are inherently with poor visible light modulation, or for planar structures that need to improve visible light transmission at low τ .

18. The supplementary Figure 31 has been added as follow:

Supplementary Figure 31: **Simulation for energy-saving performance in buildings a** The building model used in the simulation. **b** Detailed total energy consumption (include cooling, heating, and lighting) in the building. A house model was built to simulate the energy-saving with installing different windows by EnergyPlus, the window-to-wall ratio is 0.3. Here, the τ_c was set at 24 °C to better demonstrate the potential of the rolled-up SW in the future energy conditioning of the house. The calculated results clearly show that the rolled-up SWs provide energy savings in 7 cities whatever the climates are. The building model and simulation parameters list in supplementary Table 5 and 6.

19. The supplementary Figure 32 has been added as follow:

Supplementary Figure 32: **Wafer-scale VO₂ SW grown on a 4-inch quartz substrate.**
a Comparison of patterned VO₂ NM wafer (4-inch) with unpatterned VO₂ NM wafer (4-inch) in transparency. **b** Photograph of patterned VO₂ NM wafer (4-inch). **c**, Photograph of unpatterned VO₂ NM wafer (4-inch).

20. The supplementary table 5 has been added as follow:

Supplementary Table 5 Building parameters used in the EnergyPlus simulation.

Building type	Small office building
Number of Floors	1
Toral Floor Area	20m × 10m
Average Window-to-Wall Ratio	30%
Temperature setting point for HVAC control	Below 21°C for heating/Above 24°C for cooling
Setpoint for lighting on	1000 lux

21. The supplementary table 6 has been added as follow:

Supplementary Table 6 Optical information of the window used in the simulation.

	Normal window	VO ₂ NM		Rolled-up SW	
States	-	Cold state	Hot state	Cold state	Hot state

($\tau_c=24^\circ\text{C}$)					
Solar transmittance	0.775	0.3527	0.2347	0.724	0.2347
Front side solar reflectance	0.071	0.3112	0.2691	0.219	0.2691
Back side solar reflectance	0.071	0.2357	0.1742	0.239	0.1742
Visible transmittance	0.881	0.3247	0.3038	0.722	0.3038
Front side visible reflectance	0.08	0.194	0.1735	0.181	0.1735
Backside visible reflectance	0.08	0.0829	0.0893	0.16	0.0893
Infrared transmittance	0	0	0	0	0
Front side infrared emissivity	0.84	0.823	0.706	0.792	0.706
Back side infrared emissivity	0.84	0.84	0.84	0.84	0.84

22. The supplementary table 7 has been added as follow:

Supplementary Table 7 The annual energy saving of different thermochromic SW in Hong Kong, Shanghai and Albuquerque, respectively.

Thermochromic SW (window-to-wall ratio)	Hong Kong		Shanghai		Albuquerque	
	$E_c(\text{MJ}/\text{m}^2)$	$E_{\text{saving}}(\%)$	$E_c(\text{MJ}/\text{m}^2)$	$E_{\text{saving}}(\%)$	$E_c(\text{MJ}/\text{m}^2)$	$E_{\text{saving}}(\%)$
Rolled-up SW (0.3)	48.05	10.56	26.10	5.69	71.05	13.08
Perovskite/Low E glass (0.375)	27.65	9.1				
Perovskite/Hydrogel (0.385)	25.65	9.2				

Phase change polymer (0.5)		34.09	13.8				
Hydrogel (0.5)		150					
Liquid hydrogel (0.5)	1 mm			14.73	10.7		
	1 cm			25.68	19.2		
VO₂ radiation cooling (0.33)							6.5

23. The supplementary note 5 has been added as follow:

Supplementary Note 5: **Simulation ΔT_{sol} as a function of K and L/L^* .**

The transmittance simulation was performed using commercial software Essential Macleod (Thin Film Center Inc, USA) based on the multilayer optical theory with a transfer-matrix method. The optical constants (n, k) of VO₂ nanomembrane in low and high τ were taken from literature in the range of 350-2500 nm⁵⁴. The refractive index of the quartz substrate was 1.5. The thickness of VO₂ NM was fixed at 100 nm when simulating the spectra transmittance $T(\lambda)$ at low and high τ .

At high τ , the rolled-up SW maintained flat on the quartz substrate, and its transmittance $\bar{T}_{sol,high \tau,rolled-up}$ is approximately equal to the transmittance of the VO₂ NM $\bar{T}_{sol,high \tau,NM}$:

$$\bar{T}_{sol,high \tau,rolled-up} = \bar{T}_{sol,high \tau,NM} \quad (21)$$

At low τ , the rolled-up SW shifts to rolled state, and the relationship between the transmittances and the geometric parameters can be approximately described as:

$$\bar{T}_{sol,low \tau,rolled-up} = \bar{T}_{sol,low \tau,NM} \cdot [1 - L \cdot (L - 1/k)/L^*{}^2] + L \cdot (L - 1/k)/L^*{}^2 \quad (22)$$

As mentioned in equation 20, solar modulation ability of rolled-up SW can be calculated:

$$\Delta \bar{T}_{sol,rolled-up} = \bar{T}_{sol,low \tau,rolled-up} - \bar{T}_{sol,high \tau,rolled-up} \quad (23)$$

REVIEWERS' COMMENTS

Reviewer #1 (Remarks to the Author):

The authors have made careful revision according to the comments of all reviewers, and the manuscript has been greatly improved. I have no further questions.

Reviewer #2 (Remarks to the Author):

The authors provided a very elaborate and quite satisfactory revision of the manuscript following my comments. Therefore, I recommend publication of the revised version.

Reviewer #3 (Remarks to the Author):

The authors have updated the manuscript, performed new experiments, and added additional data as per the comments.

I am happy with the updates and this version of the manuscript.

Reviewer #4 (Remarks to the Author):

The authors have carefully revised the manuscript and clarified all the concerns, thus the paper should be accepted as it.